# JAFAR: Jack up Any Feature at Any Resolution

[*]**Paul Couairon**[1,2]    [*]**Loïck Chambon**[1,3]    **Louis Serrano**[1]
**Jean-Emmanuel Haugeard**[2]    **Matthieu Cord**[1,3]    **Nicolas Thome**[1,4]
[1]Sorbonne Université, CNRS, ISIR, F-75005 Paris, France  [2]Thales, TSGF, cortAIx Labs, France
[3]Valeo.ai  [4]Institut Universitaire de France (IUF)

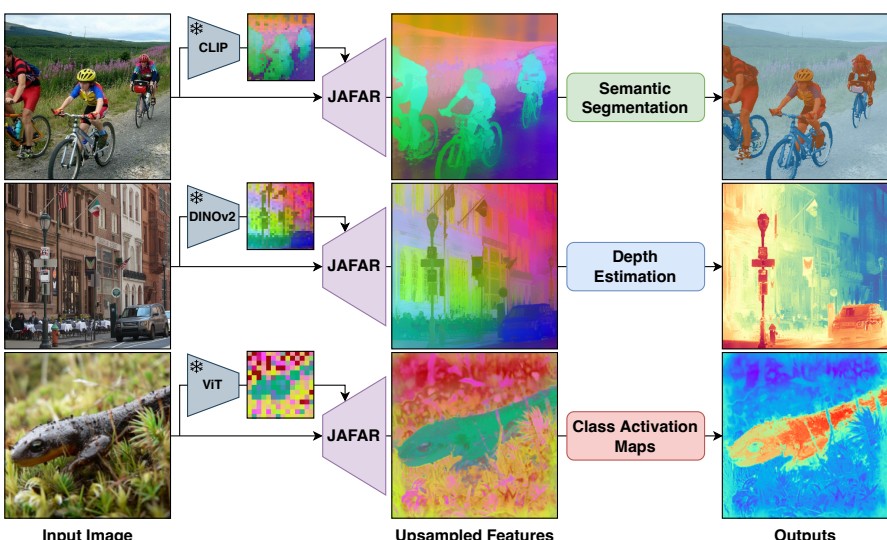

Figure 1: JAFAR upsamples features from any foundation vision encoder to any image resolution, using the input image as high-resolution guidance. It generates sharp, boundary-aligned feature maps and serves as a versatile drop-in module for a variety of downstream tasks, including semantic segmentation, open-vocabulary segmentation, depth estimation, CAM evaluation, and bird's-eye-view segmentation—consistently enhancing performance.

## Abstract

Foundation Vision Encoders have become essential for a wide range of dense vision tasks. However, their low-resolution spatial feature outputs necessitate feature upsampling to produce the high-resolution modalities required for downstream tasks. In this work, we introduce JAFAR, a lightweight and flexible feature upsampler that enhances the spatial resolution of visual features from any Foundation Vision Encoder to an arbitrary target resolution. JAFAR employs an attention-based module designed to promote semantic alignment between high-resolution queries, derived from low-level image features, and semantically enriched low-resolution keys, using Spatial Feature Transform (SFT) modulation. Notably, despite the absence of high-resolution supervision, we demonstrate that learning at low upsampling ratios and resolutions generalizes remarkably well to significantly higher output scales. Extensive experiments show that JAFAR effectively recovers fine-grained spatial details and consistently outperforms existing feature upsampling methods across a diverse set of downstream tasks.

**Project page**: https://jafar-upsampler.github.io

39th Conference on Neural Information Processing Systems (NeurIPS 2025).

# 1 Introduction

Foundation vision encoders—whether trained with language supervision [1, 2, 3, 4, 5] or purely on visual data [6, 7, 8]—have become core components of modern computer vision pipelines. Vision-language models excel at tasks requiring generalization, such as zero-shot classification and open-vocabulary segmentation [9, 10]. In contrast, image-only models, which focus on visual structure, often outperform in dense prediction tasks that demand fine-grained spatial reasoning, including semantic segmentation, depth estimation, object discovery, and point tracking [11, 12, 13].

To handle high-resolution inputs and large-scale training, foundation vision encoders typically downsample spatial information aggressively—by a factor of $14\times$ to $16\times$—yielding semantically rich but spatially coarse feature maps. This compression introduces a bottleneck for downstream tasks that require pixel-level accuracy. As a result, downstream pipelines [14, 15, 11, 16, 17] often rely on interpolation or dedicated modules [18, 19] designed to produce high-resolution outputs.

Several strategies have been explored to overcome this bottleneck, but each comes with trade-offs in efficiency and output quality. A straightforward solution is to apply training-free interpolation methods, such as bilinear upsampling. While computationally efficient, these direct interpolations—relying solely on low-resolution feature maps—fail to leverage information from the original high-resolution image, often resulting in blurry outputs. Alternatively, one can upsample the input image prior to encoding to increase feature resolution. However, this approach significantly increases computational cost due to the quadratic complexity of self-attention—common in foundation models—and may introduce artifacts in the feature maps, ultimately degrading performance [20, 21].

Focusing specifically on a target downstream task, [22, 23, 24, 25, 26] learn feature upsamplers using high-resolution supervision from task-specific labels. While generally lightweight, these upsamplers depend on labeled data tied to the end application, which limits their generalization and may bias the learned features toward optimizing task-specific losses. To address this, recent methods such as LiFT [27] and FeatUp [28] adopt task-agnostic training objectives. LiFT is trained to perform $2\times$ upsampling by regressing feature maps extracted from images at twice the input resolution. However, its convolution-based architecture is limited to fixed $2\times$ scaling, restricting its flexibility for arbitrary output resolutions. FeatUp, in contrast, uses augmented views and self-reconstruction to support higher upsampling ratios. Yet, its Joint Bilateral Upsampling (JBU) variant suffers from over-smoothed outputs, while its implicit variant requires training the upsampler for each image, making it impractical in real-world scenarios.

In this paper, we introduce a feature upsampler designed to satisfy the following criteria: *(i)* a task-agnostic training objective, *(ii)* support for arbitrary output resolutions, *(iii)* compatibility with any vision encoder, and *(iv)* minimal computational overhead at inference time.

To enable upsampling to arbitrary target resolutions, we formulate our approach as a global interpolation mechanism using a cross-attention block. The success of this attention-based method depends critically on achieving strong semantic alignment between the queries and keys. In JAFAR, we construct these representations asymmetrically (see Fig. 2): the queries retain high-resolution, low-level details such as color and texture, while the keys are hybrid features that combine high-level semantics with spatial cues. We find that enriching the keys with low-level information significantly improves query-key alignment and enhances generalization to unseen output resolutions.

Additionally, we propose a simple training objective similar to [27], but without being constrained to a fixed upsampling factor. Notably, we find that training on low upsampling factors at low resolutions (e.g., $8 \times 8 \to 32 \times 32$) is sufficient to generalize effectively to much larger scales (e.g., $32 \times 32 \to 448 \times 448$) while keeping memory requirements low during training, unlike training directly at higher resolutions and factors. Our contributions can be summarized as follows:

- We introduce JAFAR, a novel lightweight attention-based feature upsampler that naturally supports upsampling to arbitrary resolutions. It explicitly promotes spatial alignment between high-resolution queries extracted from low-level image features and semantically enriched low-resolution keys.
- We enforce this alignment by computing both queries and keys from the same input features, and injecting semantic information from the encoder's deep features via spatial feature modulation. This design enables precise fusion of spatial detail and semantic context without reliance on external supervision.

- We propose a highly efficient, task-agnostic training objective that requires no high-resolution supervision signal. Remarkably, we show that training at low resolutions and low upsampling ratios generalizes robustly to significantly higher output scales.

- We demonstrate that the combination of our architecture and training objective yields substantial performance gains across a variety of downstream tasks. When used as a drop-in module, JAFAR consistently outperforms existing upsampling methods by a wide margin.

## 2  Related Work

**Feature Upsampling**    Feature upsampling aims to increase the spatial resolution of intermediate feature maps within deep networks—analogous to image upsampling, but performed in a latent space. This process is essential for dense prediction tasks such as segmentation and depth estimation, where fine spatial detail is critical. Traditional interpolation techniques, such as bilinear, spline, or Lanczos [29, 30, 31, 32], provide simple and efficient baselines but do not adapt to the underlying content. Recent neural methods improve on static approaches by learning to reconstruct high-resolution features from data. These methods fall into two categories: task-dependent, trained with downstream labels supervision, and task-agnostic, trained independently of the end task. For example, CARAFE [22] and DySample [24] predict content-aware kernels or dynamic sampling positions. SAPA [23] and ReSFU [25] exploit a similarity based approach to refine spatial semantics. However, task-specific reliance on labels limits generalization. Recent task-agnostic methods like LiFT [27] and FeatUp [28] remove this dependency. LiFT introduces a CNN module trained with a simple fixed scale training objective, while FeatUp relies on a complex multi-loss objective which makes training difficult to tune in practice. Moreover, it requires training both an upsampler and a downsampler, adding unnecessary computational overhead. Notably, its best performance is achieved through per-image optimization, further limiting its practicality. In contrast, JAFAR provides a scalable task-agnostic framework that generalizes across resolutions without complex pipelines or per-image optimization, showing strong performance even when trained on small upsampling factors at low resolution.

**Architectural Design for Upsampling Modules**    Upsampling modules architectures vary from fixed-scale decoders to continuous resolution predictors. LiFT [27] relies on a lightweight CNN module trained to upsample by a fixed factor, making further scaling dependent on iterative use which leads to performance degradation or additional interpolation steps. FeatUp [28] introduces two architectural variants: a fast Joint Bilateral Upsampler (JBU) and a more accurate implicit network allowing continuous querying. While the implicit model yields superior results, it suffers from significant inference latency due to per-image optimization. JBU, on the other hand, trades expressivity for scalability, stacking multiple ×2 stages to achieve higher upsampling ratios. Attention-based designs, as in SAPA [23] and ReSFU [25], offer increased flexibility by modeling affinities between features across scales. These methods exploit spatial similarities to reconstruct high-resolution maps. JAFAR innovates by unifying low- and high-resolution streams: it aligns high-resolution queries and low-resolution keys using shared low-level features while enriching the representation with additional semantic cues. This design maintains spatial alignment and expressivity even at large upsampling ratios, offering a robust and scalable architecture for feature reconstruction.

**Semantic Guidance and Feature Modulation**    Feature modulation techniques modulate features using conditioning information, thereby enabling spatially or semantically guided transformations. Early forms such as Conditional BatchNorm [33], AdaIN [34], and FiLM [35] apply learned scale ($\gamma$) and shift ($\beta$) parameters per channel, derived from global conditioning signals. These methods are effective for tasks involving global transformations like style transfer or classification. However, their spatial invariance limits expressiveness in tasks requiring spatial sensitivity. SPADE [36] and SFT [37] address this limitation by computing $\gamma$ and $\beta$ as full-resolution maps conditioned on dense inputs like segmentation masks. This spatial modulation enhances expressivity by enabling unique adjustments at each feature location. It can be interpreted as a parameterized, learned recombination of feature channels, analogous to a $1 \times 1$ convolution but extended with spatially varying weights. In JAFAR, feature modulation is used not only to shift feature distributions but also to inject semantics directly into the upsampling pipeline. This provides richer linear combinations of features, improving generalization and spatial fidelity without the need for per-image optimization [28].

# 3 JAFAR

JAFAR is a feature upsampler that uses the input image as high-resolution guidance to reconstruct dense feature maps. To support upsampling to arbitrary target resolutions, we formulate the method as a global interpolation mechanism based on cross-attention. The effectiveness of this attention-based approach hinges on achieving strong semantic alignment between the queries $Q$ and the keys $K$. In JAFAR, we construct the query and key representations asymmetrically. The queries retain high-resolution, low-level details such as color and texture, while the keys are designed as hybrid representations that combine high-level semantics with low-level spatial cues. We find that enriching the keys with low-level information significantly improves query-key alignment and enhances generalization to unseen output resolutions.

## 3.1 Architecture

The overall flow of our architecture is illustrated in Fig. 2. JAFAR takes as input a high-resolution image $I \in \mathbb{R}^{3 \times H \times W}$ and a low-resolution feature map $F_{lr} = f(I) \in \mathbb{R}^{C \times h_k \times w_k}$, extracted from a frozen vision encoder $f$. The image $I$ is first projected into a higher-dimensional space and processed by a lightweight encoder $E_\theta$ to obtain an intermediate representation $I_E = E_\theta(I) \in \mathbb{R}^{d \times H \times W}$, further enriched with RoPE positional embeddings [38].

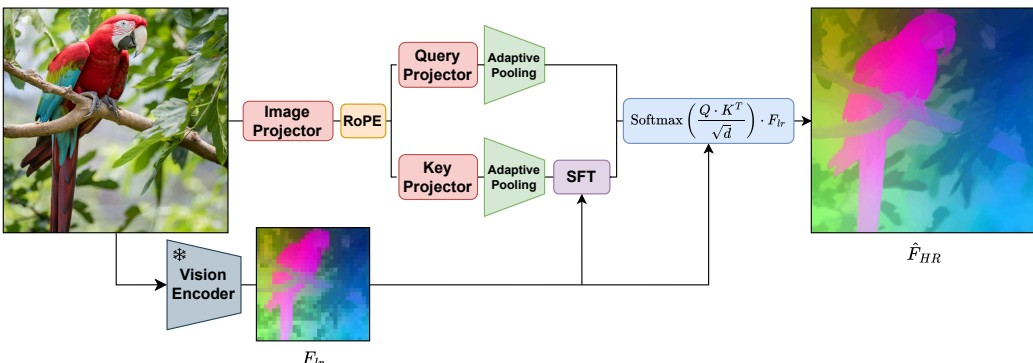

Figure 2: **Overview of JAFAR.** To construct the upsampling kernel, queries and keys are derived from a shared image representation. Queries are downsampled to match the target output resolution, while keys are downsampled to align with the spatial resolution of the vision encoder's features. Keys are then semantically enriched via SFT modulation to promote semantic alignment between queries and keys. The resulting kernel is then used to interpolate features from the foundation vision encoder.

Query features $Q \in \mathbb{R}^{d \times h_q \times w_q}$ are derived by passing the image representation $I_E$ through a small query encoder, producing $I_Q$, followed by adaptive average pooling to reach the target resolution ($h_q \times w_q$). Key features $K \in \mathbb{R}^{d \times h_k \times w_k}$ are similarly obtained by encoding $I_E$ to $I_K$ and downsampling it to match the spatial resolution of the semantic features $F_{lr}$. These semantic features provide modulation parameters that inject high-level information into the keys. A cross-attention mechanism then enables the queries $Q$ to attend to the keys $K$ by computing an attention map:

$$A = \text{Softmax}\left(\frac{Q \cdot K^\top}{\sqrt{d}}\right), \tag{1}$$

which is then used to interpolate the low-resolution feature map $F_{lr}$ and produce the upsampled output features $\hat{F}_{HR} = A \cdot F_{lr} \in \mathbb{R}^{C \times h_q \times w_q}$. The resulting representation preserves fine-grained spatial details while remaining semantically consistent with the input image. We provide a detailed description of each of the main components of the architecture below.

**Query Branch** Directly aligning high-resolution, low-level queries with high-level semantic keys often results in weak or noisy attention, as the disparity in abstraction levels limits meaningful interactions. To overcome this challenge, we apply adaptive average pooling to downsample the intermediate representation $I_Q$ and generate the query features $Q$. This operation, performed exclusively during

training, reduces the spatial resolution of the queries while aggregating local context into region-level descriptors. As a result, the downsampled queries are more semantically aligned with the keys, less susceptible to pixel-level noise, and computationally more efficient due to the reduced number of tokens. These effects collectively make query downsampling an effective strategy for bridging the gap between fine-grained visual details and abstract semantic representations, promoting more stable and scalable cross-scale attention. Importantly, because downsampling is only applied during training, the model maintains its capacity to generate high-resolution outputs during inference.

**Key Branch**   Relying exclusively on low-resolution features from the vision encoder to construct keys leads to poor generalization and noticeable artifacts, primarily due to an abstraction gap between these coarse features and the fine-grained queries. As demonstrated in Sec. 4, this mismatch results in inconsistent alignment across resolutions. To address this issue, we construct hybrid key representations that retain structural alignment with the queries while incorporating the semantic richness of the vision encoder. This is achieved by encoding the intermediate representation $I_E$ to produce $I_K$, which is then downsampled to match the spatial resolution of the encoder's feature map to produce preliminary keys $\tilde{K}$. These are further modulated using the vision encoder feature map $F_{lr} \in \mathbb{R}^{C \times h_k \times w_k}$ through a spatial semantic feature modulation inspired by [36, 37]:

$$K = \gamma_F \odot \tilde{K} + \beta_F, \tag{2}$$

where $\gamma_F, \beta_F \in \mathbb{R}^{d \times h_k \times w_k}$ are spatially varying parameters obtained via linear projections from $F_{lr}$. This adaptive, feature-wise modulation enriches the keys with localized semantic context, enhancing both spatial and semantic alignment and supporting more faithful and generalizable upsampling across resolutions.

**Similarity Based Upsampling**   To perform upsampling, we use a simplified attention mechanism where attention weights are computed via a scaled dot product between queries and semantically modulated keys. Crucially, both queries and keys have been enriched with relative positional embeddings using RoPE [38], which introduces an inductive bias that captures spatial relationships between queries and keys. This positional encoding allows us to entirely bypass the arbitrary selection of neighboring keys for each query, a common heuristic in prior similarity-based methods such as [23, 25]. Without this positional grounding, the attention mechanism lacks spatial awareness and generalizes poorly to unseen resolutions. In practice, we use multiple attention heads to increase expressivity and average the resulting attention weights across heads after applying softmax. The resulting attention map $A$ is then used to interpolate the low-resolution encoder features $F_{lr}$ via a simple matrix product: $\hat{F}_{HR} = A \cdot F_{lr}$. By avoiding a learned value projection, we preserve the original feature content and enable a resolution-agnostic design that generalizes reliably across scales.

### 3.2   Training Pipeline

Learning to upsample high-resolution features without access to ground-truth supervision poses a natural challenge: how can a model learn to produce sharp high-resolution features (e.g., $448 \times 448$) when only low-resolution features are available (e.g., $32 \times 32$)? Thanks to JAFAR's architectural design, the model can be trained with a simple objective at a low target resolution without requiring supervision at the original image size, yet it still generalizes effectively to much higher upsampling ratios during inference.

**Training with Multi-Resolution Views**   To enable this, we introduce a fully annotation-free training scheme that relies only on multi-resolution views of the same image, easily obtained through standard downsampling. Given a high-resolution image $I_{HR} \in \mathbb{R}^{3 \times H \times W}$, we generate a downsampled version $I_{LR} \in \mathbb{R}^{3 \times \lfloor \frac{H}{\delta} \rfloor \times \lfloor \frac{W}{\delta} \rfloor}$ using a randomly sampled factor $\delta \in [2, 4]$. Both images are passed through the frozen vision encoder $f$, producing two feature maps: $F_{hr} = f(I_{HR}) \in \mathbb{R}^{C \times h \times w}$ and $F_{lr} = f(I_{LR}) \in \mathbb{R}^{C \times \lfloor \frac{h}{\delta} \rfloor \times \lfloor \frac{w}{\delta} \rfloor}$, respectively. JAFAR then takes $I_{HR}$ and $F_{lr}$ as input to predict an upsampled feature map $\hat{F}_{hr}$. The predicted output is aligned with the target $F_{hr}$ using a simple alignment loss, which combines cosine similarity and L2 distance [20]:

$$\mathcal{L}(\hat{F}_{hr}, F_{hr}) = 1 - \cos(\hat{F}_{hr}, F_{hr}) + ||\hat{F}_{hr} - F_{hr}||_2. \tag{3}$$

Notably, during training, JAFAR is only exposed to moderate upsampling factors (up to $4\times$), yet it generalizes remarkably well to much higher resolutions at test time—without access to any ground-truth high-resolution features.

**How is it different from LiFT?** While our training objective is similar to that of LiFT, our approach demonstrates significantly greater capability, as shown in Tabs. 1 and 2. LiFT relies on a CNN-based architecture and is trained for fixed 2× upsampling at two predefined resolutions. As a result, it struggles to extrapolate beyond that setting without additional heuristics such as iterative upsampling or bilinear fallback. In contrast, JAFAR maintains a resolution-agnostic design which generalizes to much higher upsampling factors using this similar simple training setup.

## 4 Experiments

### 4.1 Experimental Setup

In our experiments, we train JAFAR on a single NVIDIA A100 on ImageNet training set for 100K steps using AdamW optimizer [39], with a learning rate of $2e{-}4$ and a batch size of 4. The input images fed into the foundation vision encoder are resized to $448 \times 448$, producing high-resolution target feature maps $F_{hr}$ of size $32 \times 32$ or $28 \times 28$, depending on the encoder's patch size (14 or 16). For improved training efficiency, the guidance image input to JAFAR is downsampled to $224 \times 224$.

### 4.2 Qualitative Comparisons

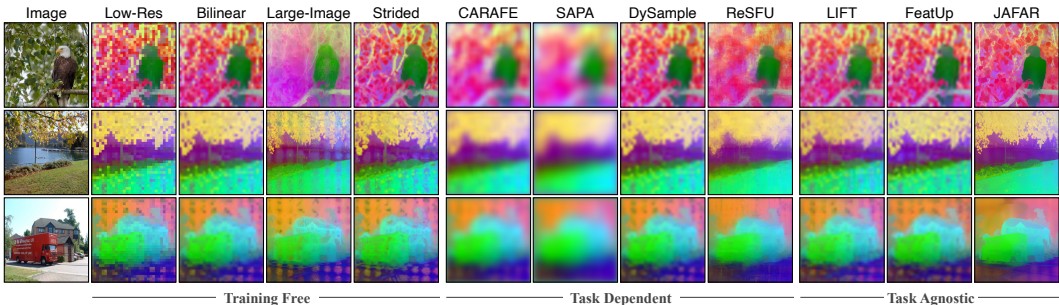

Figure 3: **PCA Feature Visualization.** DINOv2 ViT-S/14 features at $32 \times 32$ resolution from the ImageNet validation set are upsampled to $448 \times 448$. Baseline methods—whether training-free, task-dependent, or task-agnostic—introduce varying levels of blurriness and artifacts. Besides being task-agnostic, JAFAR produces sharp, content-aware feature maps with fewer artifacts.

To qualitatively evaluate the upsampled feature maps produced by various baselines, we project all features onto a shared 3-dimensional PCA basis, mapping them into a common RGB space. As shown in Figs. 3 and 5, the low-resolution features—due to the spatial compression imposed by the vision encoder's patch size—reveal large, blocky regions that capture semantic content but fail to preserve fine image geometry, object boundaries, or shape details. Bilinear upsampling, which interpolates features without considering image content, yields blurry output feature maps that preserve positional embeddings artifacts without adding meaningful detail. While methods like Large-Image and Strided preserve sharpness, their outputs are noisier and less coherent than JAFAR's. Furthermore, they are more computationally demanding, as they require the vision encoder to process a larger number of patches (see Tab. 12). JAFAR shows a clear qualitative advantage over all baselines, consistently producing sharp features that accurately capture image structure. It is also the only task-agnostic method that effectively suppresses artifacts from positional embeddings in the low-resolution features.

### 4.3 Transfer on Downstream Tasks

Since upsampled features are expected to provide a richer signal for downstream tasks, we evaluate their effectiveness on two benchmarks: linear-probing semantic segmentation and depth estimation, using DINOv2 ViT-S/14 as the foundation vision encoder. For the Large-Image and Strided baselines, upsampling is performed during the encoder's forward pass and followed by bilinear interpolation to reach the target output resolution. For task-agnostic upsamplers such as LiFT, FeatUp, and JAFAR, we pre-train the upsampling module on the corresponding backbone, then freeze it and apply it after feature extraction. The linear probe is trained independently of the upsampler. For task-dependent

methods, including CARAFE, SAPA, ReSFu, and DySample, we jointly train both the upsampler and the linear probe on each dataset and task. All experiments (except Large-Image) use input images of resolution $448 \times 448$, with target labels at the same resolution.

### 4.3.1 Semantic Segmentation

For semantic segmentation, we train a linear projection head to predict coarse class labels using a cross-entropy loss across several benchmark datasets: COCO-Stuff [40] (27 classes), ADE20K [41] (150 classes), Pascal VOC [42] (21 classes including background), and Cityscapes [43] (27 classes). The linear layer is trained for 5 epochs on COCO-Stuff and 20 epochs on the remaining datasets, using a batch size of 4. Performance is evaluated on the respective validation sets using mean Intersection-over-Union (mIoU) and pixel-wise accuracy.

Table 1: **Linear Probing on Downstream Tasks.** JAFAR consistently outperforms other baselines across all segmentation benchmarks while reaching competitive depth metrics without being optimized on a specific downstream task.

| DINOv2-ViT-S/14 | Semantic Segmentation | | | | | | | | Depth Estimation | |
| --- | --- | --- | --- | --- | --- | --- | --- | --- | --- | --- |
| | COCO | | VOC | | ADE20K | | Cityscapes | | COCO | |
| | mIoU ($\uparrow$) | Acc ($\uparrow$) | mIoU ($\uparrow$) | Acc ($\uparrow$) | mIoU ($\uparrow$) | Acc ($\uparrow$) | mIoU ($\uparrow$) | Acc ($\uparrow$) | $\delta_1$ ($\uparrow$) | RMSE ($\downarrow$) |
| *Training-free* | | | | | | | | | | |
| Nearest | 56.17 | 76.97 | 76.41 | 93.80 | 37.27 | 71.91 | 54.05 | 90.36 | 58.08 | 0.70 |
| Bilinear | 59.03 | 79.07 | 80.70 | 95.17 | 39.23 | 73.69 | 59.37 | 92.47 | 59.92 | 0.66 |
| Large Image (x8) | – | – | 56.94 | 88.60 | 26.42 | 66.39 | 47.72 | 92.49 | – | – |
| Strided | 55.93 | 77.40 | 75.88 | 93.94 | 36.15 | 72.08 | 59.26 | 92.57 | 56.98 | 0.70 |
| *Task-Dependent* | | | | | | | | | | |
| CARAFE [22] | 59.73 | 79.65 | 80.26 | 95.14 | 38.30 | 73.42 | 56.05 | 91.83 | 61.42 | 0.64 |
| SAPA [23] | 57.77 | 78.28 | 77.02 | 94.07 | 35.87 | 71.85 | 50.12 | 90.02 | 60.34 | 0.67 |
| DySample [24] | 59.50 | 79.42 | 81.62 | 95.48 | 38.99 | 73.62 | 59.71 | 92.69 | 61.25 | 0.64 |
| ReSFU [25] | 60.08 | 79.84 | 80.30 | 95.05 | 38.91 | 73.93 | 55.53 | 91.62 | **66.14** | **0.56** |
| *Task-Agnostic* | | | | | | | | | | |
| FeatUp [28] | 60.10 | 79.95 | 81.08 | 95.32 | 38.82 | 73.74 | 56.06 | 91.86 | 61.69 | 0.64 |
| LIFT [27] | 58.18 | 78.95 | 78.06 | 94.62 | 38.73 | 73.69 | 58.75 | 92.60 | 57.04 | 0.70 |
| JAFAR | **60.78** | **80.47** | **84.44** | **96.28** | **40.49** | **74.92** | **61.47** | **93.42** | 62.18 | 0.62 |

As shown in Tab. 1, JAFAR consistently achieves the highest performance across all four semantic segmentation benchmarks, in both mIoU and accuracy. On average, JAFAR delivers a $+1.63$ mIoU improvement over the next-best method across all datasets. Compared to FeatUp, JAFAR achieves an average gain of $+2.78$ mIoU corresponding to a $+4.8\%$ gain, with a peak improvement of $+5.41$ mIoU ($+9.7\%$) on Cityscapes. Fig. 4 shows linear probe segmentation result.

### 4.3.2 Depth Estimation

For depth estimation, we follow the approach in [28] and train on pseudo-labels generated by the state-of-the-art Depth Anything V2 network [16]. We report two standard metrics from the monocular depth estimation literature: root mean square error (RMSE) and $\delta_1 < 1.25$. The $\delta_1$ metric measures the percentage of pixels where the predicted depth $y$ is within 25% of the ground-truth $y^*$, formally defined as $\delta_1 = max \left( \frac{y}{y^*}, \frac{y^*}{y} \right) < 1.25$. We train the linear probe for 5 epochs on the COCO training set, using a batch size of 4. Although JAFAR was not trained on this specific task, we observe that it reaches competitive scores, ranking second among the baselines. Notably, JAFAR outperforms both FeatUp and LiFT while also surpassing all task-dependent methods but ReSFU. Fig. 4 shows linear probe depth estimation result.

### 4.3.3 Class Activation Maps Faithfulness

Following the approach in [28], our method can be seamlessly integrated into explainability tools such as Class Activation Maps (CAMs). Despite recent advances, CAMs are still fundamentally limited by the low-resolution feature maps produced by standard vision encoders, which hinders their ability to localize fine-grained details. By upsampling the features, our method yields sharper and more informative explanations. To assess the quality of the resulting CAMs, we adopt standard evaluation metrics from the literature: Average Drop (A.D), Average Increase (A.I), Average Gain (A.G), Coherency (Coh.), and Complexity (Cplx.).

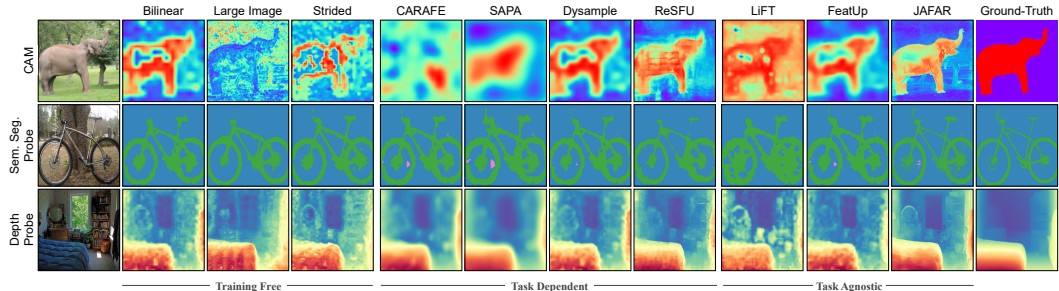

Figure 4: **Visual Comparison of Upsampler Outputs in Downstream Tasks.** JAFAR-upsampled features produce sharper outputs that align more accurately with object boundaries across various downstream tasks respectively class activations maps, semantic segmentation and depth estimation.

In particular, A.D, A.I, and A.G measure how sensitive the classifier's output is to the most salient regions of the input—an effective CAM should highlight areas that, when masked, lead to a notable change in classification confidence. Since each of these metrics captures only a single aspect of CAM quality, we also report the ADCC score—an aggregate metric proposed in [44] that provides a more holistic evaluation. Additional details are provided in Supp. A.1. As illustrated qualitatively in Fig. 4, JAFAR generates sharper and more semantically accurate CAMs compared to all baselines.

Table 2: **Grad-CAM Evaluation.** Integrating JAFAR into Grad-CAM analysis yields significantly more faithful explanations compared to baseline methods. Top three methods are highlighted as first, second, third according to **ADCC**.

| | **A.D** (↓) | **A.I** (↑) | **A.G** (↑) | **Coh.** (↑) | **Cplx.** (↓) | **ADCC** (↑) |
|---|---|---|---|---|---|---|
| *Training-free* | | | | | | |
| Bilinear | 19.0 | 18.5 | 3.4 | 88.8 | 59.9 | 61.7 (-11.6) |
| Large Image (x8) | 48.8 | 12.8 | 2.5 | 67.9 | 38.0 | 59.5 (-13.8) |
| Strided | 19.8 | 15.1 | 3.5 | 85.9 | 54.2 | 65.3 (-8.0) |
| *Task-Dependent* | | | | | | |
| CARAFE [22] | 49.9 | 4.8 | 1.0 | 66.9 | 34.9 | 59.7 (-13.6) |
| SAPA [23] | 8.5 | 32.7 | 4.1 | 96.5 | 69.6 | 55.4 (-17.9) |
| Dysample [24] | 17.8 | 20.0 | 3.8 | 90.2 | 60.7 | 61.6 (-11.7) |
| ReSFU [25] | 14.5 | 24.0 | 3.7 | 92.1 | 64.2 | 59.4 (-13.9) |
| *Task-Agnostic* | | | | | | |
| FeatUp [28] | 15.3 | 24.0 | 4.3 | 91.6 | 58.2 | 64.3 (-9.0) |
| LiFT [27] | 66.9 | 8.7 | 2.3 | 65.2 | 9.4 | 53.0 (-20.3) |
| JAFAR | 17.4 | 30.9 | 6.5 | 91.4 | 44.1 | 73.3 |

baselines. While training-free methods don't help to recover important regions, task-dependent approaches typically produce blurrier and less precise maps. Quantitative results in Tab. 2 further support this, with JAFAR achieving the highest score on the aggregate ADCC metric—outperforming the second-best method by 8 points, a relative improvement of 12.5%.

#### 4.3.4 Zero-Shot Open-Vocabulary Segmentation

We further evaluate our method on a zero-shot open-vocabulary segmentation task, following the setup from [9], where class labels from the dataset serve as textual inputs and predictions are made by selecting the class with the highest similarity score (argmax). Using a CLIP-ViT-B/16 backbone, this approach is entirely training-free, as it does not require a learned probing head. Results show that JAFAR significantly outperforms all baselines, with particularly strong improvements on Pascal VOC. Despite the increased difficulty of ADE20K, which includes 150 classes, our method still achieves the highest performance in both mIoU and accuracy. We report only FeatUp among the task-agnostic baselines, as it is the second-best performing method.

Table 3: **Zero-Shot Open-Vocabulary Evaluation.** Using MaskCLIP [9] for zero-shot open-vocabulary segmentation, JAFAR consistently improves performance, indicating strong alignment with the original features.

| | **VOC** | | **ADE20K** | | **Cityscapes** | |
|---|---|---|---|---|---|---|
| **Upsampling** | mIoU (↑) | Acc (↑) | mIoU (↑) | Acc (↑) | mIoU (↑) | Acc (↑) |
| Nearest | 24.13 | 30.80 | 9.33 | 24.65 | 19.66 | 50.27 |
| Bilinear | 27.87 | 35.27 | 11.03 | 27.78 | 21.56 | 53.21 |
| Large Image (×2) | 23.24 | 32.16 | 8.08 | 24.94 | 21.91 | 52.22 |
| FeatUp [28] | 32.27 | 39.78 | 13.03 | 33.28 | 24.76 | 60.11 |
| JAFAR | **35.70** | **44.93** | **13.61** | **33.28** | **25.26** | **61.73** |

#### 4.3.5 Bird's-Eye View Segmentation

Finally, we studied the impact of our upsampler in a complex training pipeline. The task, evaluated on nuScenes [48], takes several images taken from cameras as input and consists on outputting the bird's-eye view (BeV) segmentation map. In our setup, we used a frozen DINOv2 [7] backbone and trained the rest of the architecture, namely, the upsampler, the BeV encoder, and

Table 4: **BeV Vehicle Segmentation.** JAFAR consistently improves vehicle-IoU in complex BeV architectures, outperforming all other baselines.

| Upsampling | SimpleBeV [45] | PointBeV [46] | BeVFormer [47] |
|---|---|---|---|
| Low-Res | 31.75 | 34.89 | 33.72 |
| Bilinear | 33.67 | 36.01 | 34.18 |
| FeatUp | 33.95 | 35.38 | 34.01 |
| JAFAR | **36.59** | **37.20** | **36.54** |

the segmentation head. This task is particularly challenging, as the model must learn to map features from the image plane to the BeV plane. To ensure a fair comparison, we also trained the architecture without an upsampler, using lower-resolution input images ($224 \times 476$). We adopted the optimization hyperparameters from PointBeV [46], adjusting the batch size to 1 and training for 100 epochs. Our results show that using an upsampler consistently improves predictions, regardless of the architecture employed—SimpleBev [45], PointBeV [46], or BevFormer [47]. Notably, performance improves significantly when using JAFAR as the upsampler, with mIoU gains up to $+5$ points.

### 4.4 Ablations

To evaluate the benefit of deriving both queries and keys from a shared image encoding, we compare in Tab. 5 several strategies to obtain keys. In the Linear Projection baseline, keys are obtained by applying a linear layer to the vision encoder's low-resolution features $F_{lr}$, to match JAFAR's embedding dimension. In the Concatenation baseline, semantics is injected via a direct concatenation of $F_{lr}$ and preliminary keys $\tilde{K}$.

Table 5: **Attention mechanism ablations** with respect to key strategy and number of attention heads. Best scores per dataset are in bold and selected choices are highlighted in blue.

| Ablation Type / Setting | Semantic Segmentation | | | | | |
|---|---|---|---|---|---|---|
| | VOC | | ADE20K | | Cityscapes | |
| | mIoU (↑) | Acc (↑) | mIoU (↑) | Acc (↑) | mIoU (↑) | Acc (↑) |
| *Keys Strategy* | | | | | | |
| Linear Projection | 80.02 (-4.42) | 94.87 (-1.41) | 37.87 (-2.62) | 73.22 (-1.70) | 52.45 (-9.02) | 90.80 (-2.62) |
| Concatenation | 83.13 (-1.27) | 95.94 (-0.34) | 40.06 (-0.43) | 74.56 (-0.36) | 58.70 (-2.77) | 92.65 (-0.77) |
| w/o SFT | 83.25 (-1.19) | 95.93 (-0.35) | 39.62 (-0.87) | 74.32 (-0.60) | 56.53 (-4.94) | 92.19 (-1.23) |
| w/ SFT | **84.44** | **96.28** | **40.49** | **74.92** | **61.47** | **93.42** |
| *Attention Heads* | | | | | | |
| $n = 1$ | 84.13 (-0.31) | 96.21 (-0.07) | 40.15 (-0.34) | 74.79 (-0.13) | 60.94 (-0.53) | 93.32 (-0.10) |
| $n = 2$ | 84.27 (-0.17) | 96.27 (-0.01) | 40.42 (-0.07) | **74.95** (+0.03) | 61.19 (-0.28) | 93.42 (-0.00) |
| $n = 4$ | **84.44** | **96.28** | **40.49** | 74.92 | **61.47** | **93.42** |
| $n = 8$ | 83.82 (-0.62) | 96.13 (-0.15) | 40.07 (-0.42) | 74.20 (-0.72) | 60.56 (-0.91) | 93.33 (-0.09) |

In comparison, the Linear projection baseline shows a significant performance drop, and SFT consistently outperforms the concatenation approach. Increasing the number of attention heads up to 4 further enhances performance by producing more robust upsampling kernels through averaged post-softmax scores. Beyond this point, however, the benefits reverse: the per-head dimensionality becomes too low to support effective alignment, while the computational cost increases, ultimately degrading output quality.

## 5 Conclusion

We introduce JAFAR, a lightweight, attention-based feature upsampler designed with a simple training objective. It can upscale features from any foundation vision encoder to arbitrary output resolutions, without requiring supervision at the original image size or annotations from downstream tasks. Although task-agnostic, JAFAR outperforms prior state-of-the-art upsamplers across a variety of downstream tasks, despite not being trained specifically for them. This work lays the groundwork for a unified feature upsampler that could enable significantly more efficient architectures for dense vision tasks. Currently, the method requires training a separate upsampler for each backbone. Future

work will focus on making JAFAR backbone-independent at inference time and on further reducing feature-level artifacts to produce sharper outputs.

# 6 Acknowledgment

This work has been supported by chair VISA DEEP (ANR-20-CHIA-0022) Cluster PostGenAI@Paris (ANR-23-IACL-0007, FRANCE 2030). This work has been supported by PEPR Sharp (ANR-23-PEIA-0008, FRANCE 2030). This work was granted access to the HPC resources of IDRIS under the allocation 2025-AD011014763R1 made by GENCI.

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

# JAFAR: Jack up Any Feature at Any Resolution
## Supplementary Material

## A  Additional Details

### A.1  Evaluation

To evaluate Class Activation Maps (CAMs), we employ a frozen pre-trained ViT-B/16 model as the backbone and extract Grad-CAMs. We randomly sample 2,000 images from the ImageNet validation set for which the model produces correct predictions. For each image, we compute the gradients with respect to the predicted class, average them, and use the result to weight the corresponding activation maps. The weighted activations are then summed to produce the final CAM. These activation maps are upsampled from $14 \times 14$ to $224 \times 224$, resulting in high-resolution CAMs. For each CAM, we generate a masked version of the input image by applying a binary mask that highlights regions positively associated with the model's prediction. Formally, a masked image is obtained as $x_{\text{masked}} = x \odot \mathbb{1}_{\text{CAM}_c(x)>0}$. These masked images are then used to compute the evaluation metrics.

**Average Drop**  Average Drop (A.D) quantifies how much the model's confidence in the predicted class decreases when it is presented with the masked image instead of the full image. For a single image, the metric is defined as:

$$\textbf{A.D} = \frac{1}{N} \sum_{i=1}^{N} \frac{max(0, Y_i^c - O_i^c)}{Y_i^c} \cdot 100, \tag{4}$$

where $Y_i^c$ denotes the model's output score for class $c$ when using the full image, and $O_i^c$ denotes the score when using the masked version derived from the explanation map. The final A.D value is computed by averaging over a set of $N$ images.

**Average Increase**  Average Increase (A.I) measures how often the model's confidence in the predicted class is higher when using the masked image than when using the full image. It is defined as:

$$\textbf{A.I} = 100 \cdot \sum_{i=1}^{N} \frac{\mathbb{1}_{Y_i^c < O_i^c}}{N}, \tag{5}$$

where $Y_i^c$ is the model's output score for class $c$ when using the full image, and $O_i^c$ is the score when using the masked image based on the explanation map. The metric reflects the percentage of images where the explanation-based input yields a higher confidence score than the original image.

**Average Gain [51]**  Average Gain (A.G) quantifies the improvement in predictive confidence for the target class when using the masked image instead of the full image. It is defined as:

$$\textbf{A.G} = \frac{1}{N} \sum_{i=1}^{N} \frac{max(0, O_i^c - Y_i^c)}{1 - Y_i^c} \cdot 100, \tag{6}$$

where $Y_i^c$ is the model's output score for class $c$ on the full image, and $O_i^c$ is the score when using the masked version derived from the explanation map. This metric captures how much the explanation enhances the model's confidence, normalized by the room for improvement $(1 - Y_i^c)$.

**Coherency [44]**   A Class Activation Map should highlight all the relevant features that contribute to a model's prediction, while suppressing irrelevant ones in a coherent and consistent manner. Consequently, for a given input image $x$ and a target class $c$, the CAM should remain unchanged when the image is conditioned on the CAM itself. This self-consistency can be expressed as:

$$\text{CAM}_c(x \odot \text{CAM}_c(x)) = \text{CAM}_c(x) \tag{7}$$

where $\odot$ denotes element-wise multiplication. This condition implies that the CAM produced from the masked image should be identical to the original CAM, ensuring that the explanation is stable. Following the approach in [44], we use the Pearson Correlation Coefficient between the original CAM and the CAM obtained after masking:

$$\text{Coherency}(x) = \frac{\text{Cov}(\text{CAM}_c(x \odot \text{CAM}_c(x)), \text{CAM}_c(x))}{\sigma_{\text{CAM}_c(x \odot \text{CAM}_c(x))} \sigma_{\text{CAM}_c(x)}} \tag{8}$$

where Cov denotes the covariance and $\sigma$ the standard deviation of each CAM. Since the Pearson Correlation Coefficient ranges from $-1$ to $1$, we normalize it to the range $[0, 1]$ and express it as a percentage for interpretability. A coherency score of 100% indicates that the attribution method is fully invariant to input perturbations guided by its own explanations.

**Complexity**   In addition to ensuring that a CAM is coherent—preserving predictive features while discarding irrelevant ones—it is also desirable for the CAM to be as simple as possible. That is, it should highlight the minimal subset of pixels necessary to explain the model's prediction. To quantify this notion of simplicity, we use the $\ell_0$ norm as a proxy for the Complexity of a CAM:

$$\text{Complexity}(x) = \|\text{CAM}_c(x)\|_0, \tag{9}$$

where $\|\cdot\|_0$ counts the number of non-zero (i.e., activated) pixels. A lower Complexity score indicates that the attribution method focuses on fewer, more relevant regions, thereby producing more concise and interpretable explanations.

**ADCC [44]**   Since each individual metric captures a distinct aspect of CAM quality, we compute an aggregated evaluation metric—Average DCC (ADCC)—which combines Coherency, Complexity, and Average Drop into a single score using the harmonic mean:

$$\text{ADCC}(x) = 3 \left( \frac{1}{\text{Coherency}(x)} + \frac{1}{1 - \text{Complexity}(x)} + \frac{1}{1 - \text{A.D}(x)} \right)^{-1} \tag{10}$$

ADCC offers a unified, single-valued measure that enables direct and consistent comparison. By balancing coherency, sparsity (via low complexity), and confidence preservation (via low Average Drop), it provides a more comprehensive assessment of attribution quality.

### A.2   Baselines

- **Large Image**: For the Large Image baseline, we upsampled the original image via bilinear upsampling and use it as input to the foundation vision encoder. During evaluation on downstream tasks (see Tabs. 1 and 2), we upsample the input to the maximum ratio that fits in memory (i.e., $\times 8$), and subsequently apply bilinear upsampling to the resulting feature map to match the target output resolution. Due to the high computational cost and training time, we omit results on the COCO dataset. For efficiency, on Open-Vocabulary segmentation we limit upsampling to a $\times 2$ ratio in Tab. 3.

- **Strided**: To obtain higher-resolution feature maps, we modify the stride of the ViT backbone to produce more patches. While the stride typically equals the patch size (e.g., 14 in DINOv2), we reduce the former to 6 in our experiments corresponding to a $\times 2.3$ upsampling. We then apply bilinear upsampling to the resulting feature map to reach the desired output resolution.

- **CARAFE** [22]: For CARAFE, we use the CARAFEPack module from MMCV [52], stacking four upsampling stages with a $\times 2$ ratio each, resulting in a final feature map upsampled by a factor of $\times 16$.

- **SAPA** [23]: We adopt the default implementation from the official SAPA repository, stacking four SAPA upsampling modules, each with an upsampling factor of 2. This results in a final feature map with a total upsampling factor of $\times 16$.

- **DySample** [24]: We adopt the default implementation from the official Dysample repository, stacking four Dysample upsampling modules, each with an upsampling factor of 2. This results in a final feature map with a total upsampling factor of $\times 16$.

- **ReSFU** [25]: We use the default implementation from the official ReSFU repository, performing a direct upsampling to the target output resolution.

- **FeatUp** [28]: For FeatUp, we use the scalable JBU variant from the official FeatUp repository stacking 4 upsampling modules to achieve a total upsampling factor of $\times 16$. We re-train FeatUp using the provided training scripts. While the original paper trains FeatUp on the COCO dataset for 2,000 steps with a batch size of 4 and evaluates on the same dataset, we train it on the ImageNet training set for 50,000 steps ($25\times$ more) using the same batch size, ensuring a fair comparison across methods.

- **LiFT** [27]: For LiFT, we slightly adapt the official implementation by resizing the intermediate representations after the downsampling module to ensure compatibility with backbones using a patch size of 14 (i.e., a downsampling factor of 14). In the official code, LiFT performs a $\times 2$ upsampling and then relies on a bilinear interpolation to upsample the features to the target output resolution.

We summarize in Tab. 6 the differences between upsamplers.

Table 6: Comparison of feature upsampling methods.

| Method | Task-Agnostic | Direct Upsampling | Lightweight Inference |
|---|---|---|---|
| CARAFE - SAPA - DySample | ✗ | ✗ | ✓ |
| ReSFU | ✗ | ✓ | ✓ |
| LiFT | ✓ | ✗ | ✓ |
| FeatUp (JBU) | ✓ | ✗ | ✓ |
| FeatUp (Implicit) | ✓ | ✓ | ✗ |
| **JAFAR** | ✓ | ✓ | ✓ |

# B  Additional Visualizations

We provide in the following subsections additional comparison visualizations for upsampled feature maps Supp. B.1, class activation maps predictions Supp. B.2, depth estimation Supp. B.3 and semantic segmentation Supp. B.4.

## B.1  Feature Visualization

Fig. 5 shows additional PCA visualizations of upsampled feature maps. Starting from $32 \times 32$ feature maps extracted using a DINOv2-S/14 backbone, each is upsampled to $448 \times 448$ using different baseline methods. These baselines—whether training-free, task-dependent, or task-agnostic—tend to introduce varying degrees of blurriness and visual artifacts. In contrast, JAFAR, while remaining task-agnostic, produces sharp, content-aware features with minimal artifacts.

## B.2  Class Activation Maps

We present additional Grad-CAM visualizations based on ViT-B/16 features from the ImageNet validation set in Fig. 6. Except for the "Low-Res" column, where features remain at their original 14×14 resolution, all feature maps are upsampled to 224×224 before Grad-CAM extraction. The explainability maps generated by our upsampling approach are noticeably sharper and more accurate, exhibiting fewer artifacts compared to those from alternative methods. Notably, CARAFE and LiFT fail to produce meaningful explanations in this setting, suggesting that the training of these methods does not transfer effectively to these ViT-based features.

## B.3  Depth Estimation

Fig. 7 presents additional examples of linear probe transfer learning for depth estimation on the COCO-Stuff dataset. Feature maps of size $32 \times 32$, extracted from a DINOv2-S/14 backbone, are upsampled to $448 \times 448$ using the various baseline methods. A linear probe is then trained on

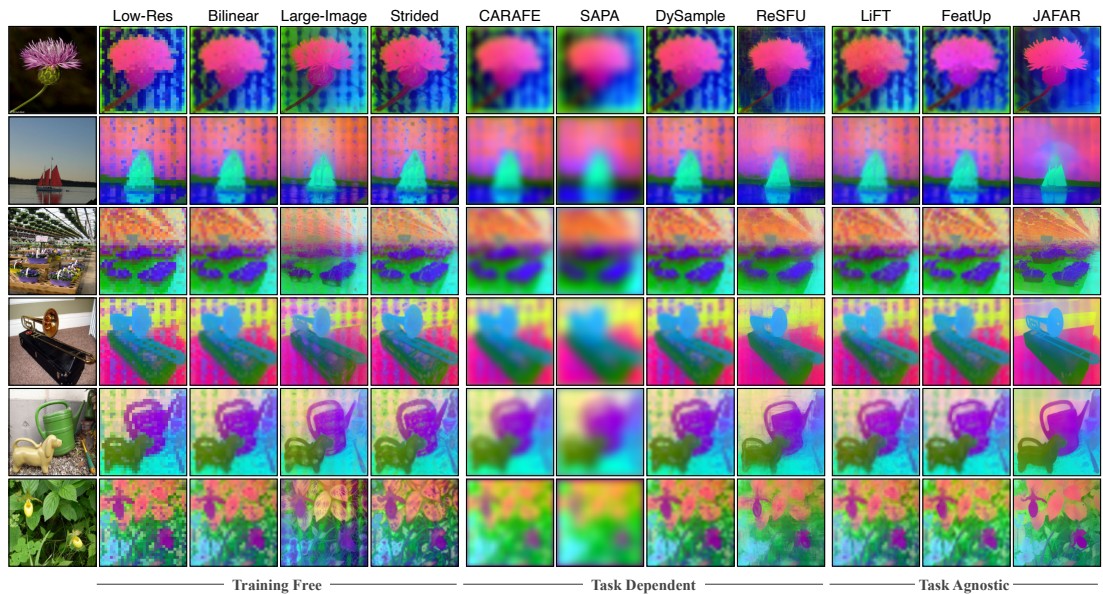

Figure 5: **PCA Feature Visualization.** DINOv2 ViT-S/14 features at $32 \times 32$ resolution from the ImageNet validation set are upsampled to $448 \times 448$.

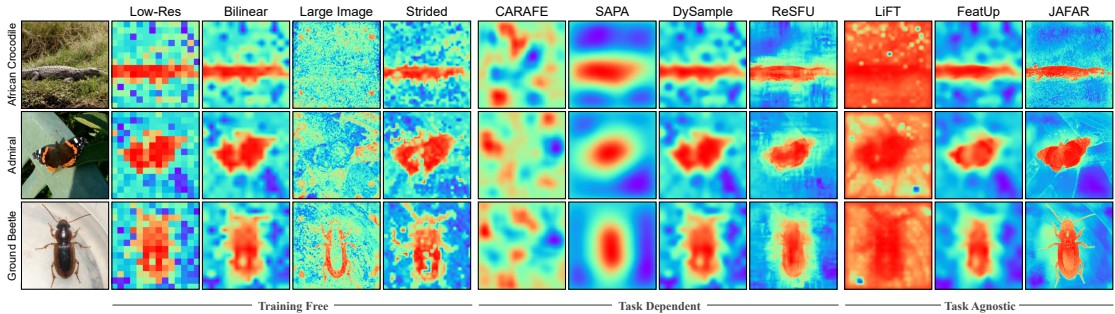

Figure 6: **Class Activation Maps comparison.**

these features to predict depth, using supervision from a Depth-AnythingV2 model. The results demonstrate that both FeatUp variants produce high-quality features well-suited for transfer learning in depth estimation tasks.

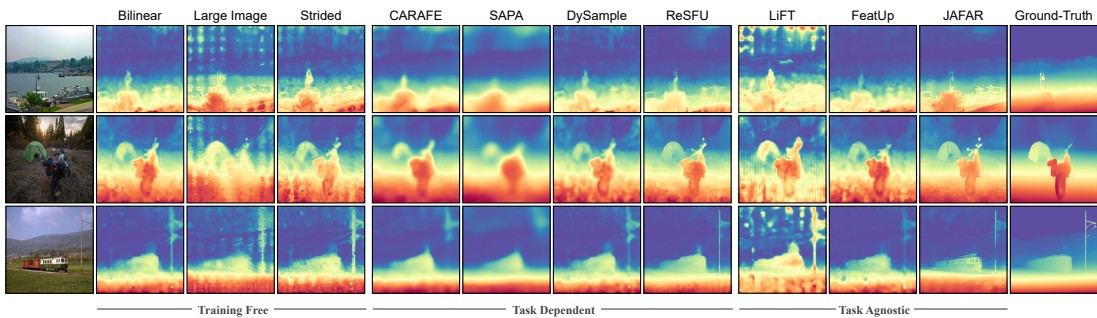

Figure 7: **Depth Estimation Visualization.**

## B.4 Semantic Segmentation

Fig. 8 presents examples of linear probe transfer learning for semantic segmentation on the COCO-Stuff dataset. Feature maps of size $32 \times 32$, extracted from a DINOv2-S/14 backbone, are upsampled to $448 \times 448$ using the various baseline methods. JAFAR produces more coherent segmentation results, offering improved delineation of both object boundaries and background regions.

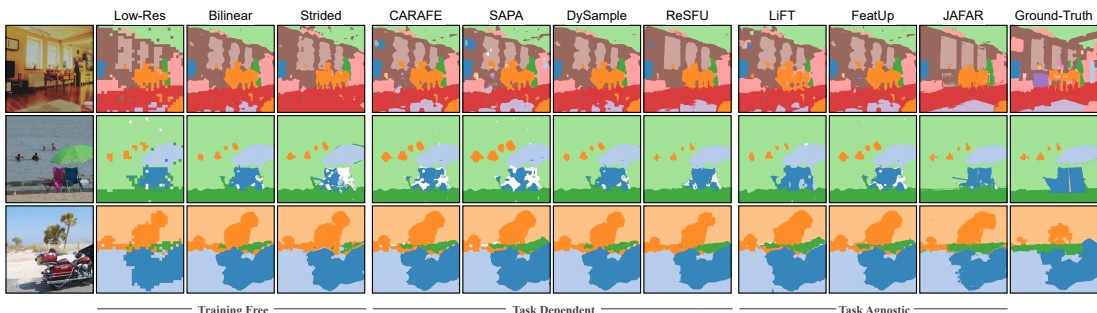

Figure 8: **Semantic Segmentation Visualization.**

## B.5 Attention Maps Visualization

To illustrate the behavior of the upsampling module, we visualize attention maps in Fig. 9. For each example, a query location is selected, and its attention distribution over the low-resolution keys is shown. The maps reveal that attention tends to concentrate on the region semantically related to the query, even though all keys are accessible. This suggests that fully global attention may not be strictly necessary, and that localized variants could provide significant runtime and memory gains. Nevertheless, we argue that the mechanism should not be overly local, since feature maps produced by vision encoders often suffer from spatial misalignments and positional artifacts. Allowing each query a broader receptive field helps counteract these inconsistencies and improves upsampling quality. Conceptually, this resembles the principle of non-local means in image denoising, where a pixel is refined using information from a wider neighborhood rather than only its immediate surroundings.

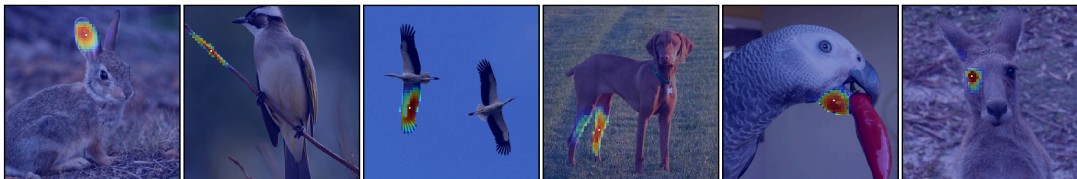

Figure 9: **Attention maps from JAFAR.** Each map shows the attention distribution of a query location (white point) over the low-resolution keys.

## C    Additionnal Comparisons With Task-Agnostic Baselines

### C.1    FeatUp & LiFT

In Tab. 1, we reported results obtained by training FeatUp and LiFT within our own codebase. To complement this evaluation, we present an additional comparison in Tabs. 8 and 10 using the official released checkpoints, FeatUp on DINOv2 ViT-S/14 and LiFT on DINO ViT-S/16, respectively.

Across all datasets and output resolutions, with the exception of ADE20K at 448 for FeatUp, JAFAR consistently delivers significant improvements in semantic segmentation performance. We also evaluate a LiFT-iterative baseline, which stacks two LiFT $\times 2$ upsamplers as described in the original LiFT paper. However, this iterative approach does not outperform the simpler method of applying a single LiFT $\times 2$ upsampler followed by bilinear interpolation.

Table 7: **Semantic Segmentation Comparison between JAFAR and FeatUp.** JAFAR is evaluated using FeatUp's original feature extractor and compared against the official JBU checkpoint on the DINOv2 ViT-S/14 backbone.

| | **224** | | | **448** | | |
| Model | ADE20K | Cityscapes | VOC | ADE20K | Cityscapes | VOC |
|---|---|---|---|---|---|---|
| Bilinear | 24.50 | 38.18 | 64.10 | 28.19 | 49.35 | 70.64 |
| FeatUp | 29.26 | 42.84 | 71.90 | **32.87** | 53.17 | 77.57 |
| JAFAR | **30.04** | **48.52** | **75.36** | 32.29 | **56.45** | **79.06** |

Table 8: **Semantic Segmentation Comparison between JAFAR and LiFT.** JAFAR is evaluated using LiFT's original feature extractor and compared against the checkpoint on the DINO ViT-S/16 backbone.

| | **224** | | | **448** | | |
| Model | ADE20K | Cityscapes | VOC | ADE20K | Cityscapes | VOC |
|---|---|---|---|---|---|---|
| Bilinear | 15.89 | 32.52 | 32.15 | 16.44 | 36.56 | 33.13 |
| LiFT | 16.97 | 36.31 | 35.30 | 16.98 | 40.01 | 35.07 |
| LiFT-iterative | 15.62 | 37.68 | 31.25 | 14.96 | 39.97 | 29.43 |
| JAFAR | **21.08** | **39.93** | **47.36** | **21.44** | **43.93** | **48.18** |

## C.2 LoftUp

We present an additional comparison with LoftUp [54], a recent baseline which relies on segmentation masks from SAM [55] during training and employs a two-stage pipeline that includes a self-distillation phase. For a fair evaluation, we tested JAFAR within LoftUp's official codebase using a DINOv2 ViT-S/14 backbone.

Table 9: **Semantic Segmentation Comparison between JAFAR and LiFT.** JAFAR is evaluated using Loftup's original feature extractor and compared against the checkpoint on the DINOv2 ViT-S/14 backbone.

| **Resolution** | | **Cityscapes** | | **COCO** | |
| | | mIoU ($\uparrow$) | Acc ($\uparrow$) | mIoU ($\uparrow$) | Acc ($\uparrow$) |
|---|---|---|---|---|---|
| $56^2$ | LoftUp | 15.30 | 75.50 | 25.33 | 54.06 |
| | JAFAR | **19.09** | **79.34** | **28.79** | **57.86** |
| | JAFAR + distillation | 18.35 | 79.05 | 28.60 | 57.49 |
| $112^2$ | LoftUp | 32.02 | 86.28 | 48.89 | 73.09 |
| | JAFAR | **34.56** | 87.19 | 50.67 | 74.56 |
| | JAFAR + distillation | 33.63 | **87.47** | **50.76** | **74.60** |
| $224^2$ | LoftUp | 50.83 | **91.55** | 59.79 | **80.04** |
| | JAFAR | 51.45 | 91.25 | 59.76 | 79.93 |
| | JAFAR + distillation | **51.84** | 91.53 | **59.90** | **80.04** |
| $448^2$ | LoftUp | **62.49** | 93.69 | 62.25 | 81.43 |
| | JAFAR | 61.49 | 93.46 | 62.02 | 81.30 |
| | JAFAR + distillation | 62.30 | **93.76** | **62.36** | **81.45** |

As shown in Tab. 9, JAFAR outperforms LoftUp at lower upsampling resolutions (56 and 112) and delivers comparable performance at higher resolutions (224 and 448). Different from LoftUp, JAFAR uses a simpler and more efficient single-stage strategy: it operates entirely at low resolution and does not rely on external annotations. Nevertheless, the self-distillation mechanism introduced in LoftUp is complementary to our approach and can be seamlessly integrated into JAFAR's pipeline. To demonstrate this, we implemented a similar distillation objective and report the results as JAFAR + distillation in Tab. 9. While the gains are minimal at lower resolutions, this enhancement provides

a clear boost at higher resolutions (224 and 448). Lastly, JAFAR is considerably more lightweight, with only 700K parameters compared to 4.3M in LoftUp.

# D    Additional Analysis

## D.1    Upsampling Ratios

To evaluate the robustness of JAFAR upsampling across different ratios, we measure its performance on a linear probing semantic segmentation task at multiple scales, including extreme ratios ($8^2 \to 896^2$). The results indicate that JAFAR consistently sustains strong performance as the upsampling ratio grows, underscoring its ability to generalize well beyond the training range.

Table 10: **JAFAR robustness** across upsampling scales on linear probing segmentation.

| Input Resolution | Dataset | Upsampler | $56^2$ | $112^2$ | $224^2$ | $448^2$ | $896^2$ |
|---|---|---|---|---|---|---|---|
| $8^2$ | Cityscapes | Bilinear | 31.87 | 31.81 | 31.90 | 31.61 | 31.92 |
| | | JAFAR | 36.54 | 36.04 | 35.71 | 35.77 | 35.54 |
| | VOC | Bilinear | 64.79 | 64.95 | 64.89 | 64.91 | 64.9 |
| | | JAFAR | 72.7 | 73.13 | 72.77 | 72.69 | 72.55 |
| $16^2$ | Cityscapes | Bilinear | 47.77 | 47.77 | 47.78 | 47.83 | 47.84 |
| | | JAFAR | 53.18 | 53.22 | 52.35 | 51.76 | 51.36 |
| | VOC | Bilinear | 75.41 | 75.42 | 75.51 | 75.50 | 75.50 |
| | | JAFAR | 80.57 | 80.76 | 80.58 | 80.36 | 80.24 |
| $32^2$ | Cityscapes | Bilinear | 59.56 | 59.53 | 59.41 | 59.54 | 59.15 |
| | | JAFAR | 59.65 | 61.39 | 61.38 | 60.86 | 60.79 |
| | VOC | Bilinear | 80.64 | 80.72 | 80.77 | 80.83 | 80.82 |
| | | JAFAR | 81.51 | 82.53 | 82.67 | 83.55 | 83.24 |

## D.2    Positional Encoding

As shown in Tab. 11, RoPE plays a critical role in maintaining consistent spatial alignment between queries and keys across varying spatial resolutions, with its benefits becoming more pronounced at higher resolutions.

Table 11: **Ablation of RoPE** on linear probing segmentation (mIoU).

| | Cityscapes | | | | VOC | | | |
|---|---|---|---|---|---|---|---|---|
| **JAFAR** | $56^2$ | $112^2$ | $224^2$ | $448^2$ | $56^2$ | $112^2$ | $224^2$ | $448^2$ |
| w/o RoPE | 18.15 | 27.06 | 34.80 | 36.09 | 35.36 | 62.38 | 67.58 | 67.22 |
| w/ RoPE | 21.41 | 35.80 | 52.42 | 60.86 | 40.41 | 72.78 | 80.57 | 83.55 |

# E    Performance

We compare in Tabs. 12 and 13 the runtime and memory usage respectively of various methods with a batch size of 1 and input resolution of 448, across multiple target resolutions. The experiments are conducted on a single A100 GPU.

Table 12: **Runtime (ms) for different models and resolutions.**

| Model | # Params (M) | $56^2$ | $112^2$ | $224^2$ | $448^2$ |
|---|---|---|---|---|---|
| *Upsamplers* | | | | | |
| FeatUp | 0.2 | 5.7 | 8.0 | 14.9 | 64.9 |
| JAFAR | 0.7 | 4.0 | 5.7 | 16.6 | 94.0 |
| LiFT | 1.2 | 0.9 | 0.9 | 1.0 | 1.5 |
| LoftUp | 4.3 | 3.8 | 8.9 | 24.5 | 145.1 |
| *Other configurations* | | | | | |
| Large Image ($\times 8$) | - | 6.2 | 34.7 | 348 | 5 558 |
| Strided (1) | - | 11.4 | 136.0 | 2 482 | 48 090 |

Table 13: **Memory usage (GB) for different models and resolutions.**

| Pass | Model | # Params (M) | $56^2$ | $112^2$ | $224^2$ | $448^2$ |
|---|---|---|---|---|---|---|
| Forward | Bilinear | 0.0 | 0.4 | 0.5 | 0.5 | 0.8 |
| | FeatUp | 0.2 | 0.6 | 0.8 | 1.6 | 4.8 |
| | JAFAR | 0.7 | 0.6 | 0.6 | 1.1 | 7.7 |
| | LoftUp | 4.3 | 0.6 | 0.7 | 1.8 | 12.3 |
| Backward | FeatUp | 0.2 | 0.7 | 0.9 | 2.1 | 7.4 |
| | JAFAR | 0.7 | 0.7 | 1.1 | 3.5 | 26.0 |
| | LoftUp | 4.3 | 0.7 | 1.3 | 4.6 | 26.5 |

In Tab. 14, we report GFLOPs and training memory comparisons for the BeV segmentation setup. JAFAR delivers $6\% - 15\%$ higher IoU while requiring only $3\% - 10\%$ more GFLOPs than the version without upsampling, underscoring the trade-off between accuracy and computation.

Table 14: **Comparison of upsampling methods on BeV segmentation pipelines.**

| Model | Metric | No Upsampling | Bilinear | FeatUp | JAFAR | LoftUp |
|---|---|---|---|---|---|---|
| SimpleBeV [45] | Memory (MiB) | 4539 | 4981 | 5305 | 5453 | 5305 |
| | GFLOPs | 594.00 | 595.10 | 595.73 | 612.24 | 621.06 |
| | mIoU | 31.75 | 33.67 | 33.95 | 36.59 | – |
| PointBeV [46] | Memory (MiB) | 1507 | 1833 | 1961 | 2521 | 2397 |
| | GFLOPs | 154.30 | 154.32 | 154.90 | 171.41 | 180.25 |
| | mIoU | 34.89 | 36.01 | 35.38 | 37.20 | – |
| BeVFormer [47] | Memory (MiB) | 2143 | 2489 | 2639 | 3135 | 2971 |
| | GFLOPs | 352.40 | 354.70 | 357.72 | 376.74 | 379.82 |
| | mIoU | 33.72 | 34.18 | 34.01 | 36.54 | – |

# F    Limitations

Although our method offers significant advantages over existing upsampling approaches, it also has certain limitations. Because our approach employs a global attention mechanism, each query attends to all keys, and the number of keys increases with the resolution of the input image processed by the foundation vision encoder. As a result, the computational and memory costs can become prohibitive when dealing with large key sets (e.g., $64 \times 64$ or higher). While a large receptive field is desirable, attending to every key is not strictly necessary. Alternative attention mechanisms with more localized receptive fields [56, 57, 58] may achieve orders-of-magnitude efficiency improvements and represent a promising direction for future work.

