# OpenReview forum: "JAFAR: Jack up Any Feature at Any Resolution"
_NeurIPS.cc/2025/Conference — NeurIPS 2025 poster_

### Official Review · Reviewer_RrSa · 2025-06-09

**Clarity:** 3
**Significance:** 3
**Originality:** 3
**Rating:** 5
**Confidence:** 4

**Summary:**

This paper proposes a novel feature upsampling module named JAFAR, which can upsample the low-resolution output features from any foundation vision encoder (such as DINOv2 or CLIP) to arbitrary high resolutions, enabling downstream pixel-level tasks such as semantic segmentation, depth estimation, and CAM visualization. Unlike existing task-dependent or fixed-ratio task-agnostic upsampling methods, JAFAR designs a heterogeneous Query-Key alignment strategy based on a cross-attention mechanism, where the Query retains the high-resolution texture information from the original image, and the Key is modulated from low-resolution semantic features via an SFT module. Experiments demonstrate the effectiveness of the proposed method.

**Questions:**

See Weakness

**Ethical Concerns:**

["NO or VERY MINOR ethics concerns only"]

**Final Justification:**

The author's response has addressed my concerns, and I raise my score to 5.

**Limitations:**

Yes

**Quality:**

3

**Strengths And Weaknesses:**

Strengths
JAFAR provides a unified and efficient feature upsampling solution for multiple dense vision tasks. It consistently improves performance without relying on downstream supervision, making it a high-quality plug-and-play module.

Compared to LiFT (which only supports 2× upsampling) or FeatUp (which has limited performance), JAFAR naturally supports arbitrary resolution alignment and is verified to maintain alignment performance even at unseen high resolutions (e.g., 448) during testing.

The paper conducts systematic evaluations across multiple tasks and benchmarks, and the results show that the proposed method achieves superior performance.

Weaknesses
Although the paper frequently emphasizes “lightweight” and “efficient,” it does not provide numbers such as FLOPs, parameters, or inference latency. Also, there is no comparison with other task-agnostic methods (like FeatUp or LiFT) in terms of GPU/CPU runtime or memory usage. Moreover, the authors do not evaluate the challenges of high-scale upsampling, where the attention computation grows quadratically with resolution. Compared to FeatUp’s layer-wise interpolation, is JAFAR more prone to out-of-memory issues under extreme upsampling ratios?

The paper does not analyze robustness to input image quality. Since JAFAR heavily depends on the original image as Query guidance, its performance may degrade when the image is blurry, occluded, or under low light. There is no robustness evaluation on low-light, blurred, JPEG-compressed, or occluded images. In other words, if degradations like occlusion or Gaussian blur are applied to the Query image, will the output deteriorate? Is the module too dependent on image quality? This is a critical issue for real-world deployment of downstream tasks.

The paper should include visualizations such as attention maps or Query-Key matching diagrams to show what the model captures from the image. Adding these would make the method more intuitive.

Although the paper claims support for 32→448 upsampling, have the authors tried more extreme cases such as 8→512 or 64→1024? It would be helpful to discuss the generalization limits of the module (e.g., patch size, image aspect ratio) or failure cases.

I am curious whether large upsampling ratios will cause Query-Key misalignment or sparse attention issues. Also, is RoPE more stable under large-scale resolutions compared to learnable positional bias or sinusoidal encoding?

---

> ### Author Rebuttal · Authors · 2025-07-28
>
> We thank the reviewer for the helpful comments and appreciate the positive feedback. Please note that, for the purpose of the rebuttal, all probing layers of the following tables were trained for 10 epochs (instead of 20, as reported in the main paper) to accelerate training while preserving fair comparison.
>
> **1. Although the paper frequently emphasizes “lightweight” and “efficient,” it does not provide numbers such as FLOPs, parameters, or inference latency. Also, there is no comparison with other task-agnostic methods (like FeatUp or LiFT) in terms of GPU/CPU runtime or memory usage.**
>
> We would like to highlight that the supplementary material includes a detailed comparison of JAFAR against other task-agnostic methods (Tab 10-11), such as FeatUp and LiFT, reporting parameter counts, inference latency, and GPU memory usage. As detailed in the supplementary material, JAFAR demonstrates runtime and memory usage comparable to FeatUp’s layer-wise interpolation strategy for upsampling resolutions up to 224×224.
>
> **2. Moreover, the authors do not evaluate the challenges of high-scale upsampling, where the attention computation grows quadratically with resolution. Compared to FeatUp’s layer-wise interpolation, is JAFAR more prone to out-of-memory issues under extreme upsampling ratios?**
>
> At higher resolutions, JAFAR incurs additional computational overhead; however, it remains scalable and stable, successfully handling feature upsampling resolutions up to 896×896 without encountering out-of-memory issues.
> That said, extreme upsampling ratios naturally lead to increased GPU memory consumption. This limitation could be alleviated by replacing global attention with more localized alternatives, such as those used in [1, 2, 3, 4]. We will make sure to explicitly include these considerations in the limitation section of the supplementary material.
>
> **3. The paper does not analyze robustness to input image quality. Since JAFAR heavily depends on the original image as Query guidance, its performance may degrade when the image is blurry, occluded, or under low light. There is no robustness evaluation on low-light, blurred, JPEG-compressed, or occluded images. In other words, if degradations like occlusion or Gaussian blur are applied to the Query image, will the output deteriorate? Is the module too dependent on image quality? This is a critical issue for real-world deployment of downstream tasks.**
>
> We thank the reviewer for highlighting this important point regarding robustness to image quality. While we did not explicitly emphasize this in the main paper, a point we will address in the revised version, we would like to clarify that, during training, we incorporated light Gaussian blur and noise augmentations to improve robustness to image degradations.
> To better assess the impact of this regularization, we include below an ablation study evaluating JAFAR’s performance under noise degradation. These results demonstrate that the model maintains reasonable performance and that the augmentation strategy contributes positively to its robustness. We appreciate the reviewer’s feedback and will make sure to integrate these insights more explicitly in the final version of the paper.
>
> |               | Cityscapes | Cityscapes + Noise |
> |---------------|------------|--------------------|
> | JAFAR w/ aug  | 60.86      | 59.00              |
> | JAFAR wo/ aug | 60.44      | 58.44              |
>
> |               | VOC   | VOC + Noise |
> |---------------|-------|-------------|
> | JAFAR w/ aug  | 83.55 | 82.98       |
> | JAFAR wo/ aug | 83.07 | 82.73       |
>
> **4. The paper should include visualizations such as attention maps or Query-Key matching diagrams to show what the model captures from the image. Adding these would make the method more intuitive.**
>
> We thank the reviewer for this valuable recommendation. We agree that visualizations of the Query-Key attention mechanism would provide a more intuitive understanding of the method, helping to visualize how the queries attend to keys for the upsampling. Following the recommendation, will include additional visualizations of the cross-attention mechanism in a dedicated supplementary section.
>
> **5. Although the paper claims support for 32→448 upsampling, have the authors tried more extreme cases such as 8→512 or 64→1024? It would be helpful to discuss the generalization limits of the module (e.g., patch size, image aspect ratio) or failure cases.**
>
> We conduct below an experiment evaluating the performance of JAFAR in a linear segmentation probing setup under various upsampling ratios including extreme scenarios. This experiment demonstrates that JAFAR maintains robust performance even for extreme upsampling cases. We thank the reviewer for this recommendation and will further discuss such cases in the supplementary material.
>
> **Input Feature Size: 8**
>
> | **Cityscapes** (mIoU) | 56 ($\times 7$) | 112 ($\times 14$) | 224 ($\times 28$) | 448 ($\times 56$) | 896 ($\times 112$) |
> |:---------------------:|:---------------:|:-----------------:|:-----------------:|:-----------------:|:------------------:|
> |        Bilinear       |      31.87      |       31.81       |       31.90       |       31.61       |        31.92       |
> |         JAFAR         |      36.54      |       36.04       |       35.71       |       35.77       |        35.54       |
>
> | **VOC** (mIoU) | 56 ($\times 7$) | 112 ($\times 14$) | 224 ($\times 28$) | 448 ($\times 56$) | 896 ($\times 112$) |
> |:--------------:|:---------------:|:-----------------:|:-----------------:|:-----------------:|:------------------:|
> |    Bilinear    |      64.79      |       64.95       |       64.89       |       64.91       |        64.9        |
> |      JAFAR     |       72.7      |       73.13       |       72.77       |       72.69       |        72.55       |
>
>
> **Input Feature Size: 16**
>
> | **Cityscapes** (mIoU) | 56 ($\times 3.5$) | 112 ($\times 7$) | 224 ($\times 14$) | 448 ($\times 28$) | 896 ($\times 56$) |
> |:---------------------:|:-----------------:|:----------------:|:-----------------:|:-----------------:|:-----------------:|
> |        Bilinear       |       47.77       |       47.77      |       47.78       |       47.83       |       47.84       |
> |         JAFAR         |       53.18       |       53.22      |       52.35       |       51.76       |       51.36       |
>
> | **VOC** (mIoU) | 56 ($\times 3.5$) | 112 ($\times 7$) | 224 ($\times 14$) | 448 ($\times 28$) | 896 ($\times 56$) |
> |:--------------:|:-----------------:|:----------------:|:-----------------:|:-----------------:|:-----------------:|
> |    Bilinear    |       75.41       |       75.42      |       75.51       |       75.50       |       75.50       |
> |      JAFAR     |       80.57       |       80.76      |       80.58       |       80.36       |       80.24       |
>
>
> **Input Feature Size: 32**
>
> | **Cityscapes** (mIoU) | 56 ($\times 1.75$) | 112 ($\times 3.5$) | 224 ($\times 7$) | 448 ($\times 14$) | 896 ($\times 28$) |
> |:---------------------:|:------------------:|:------------------:|:----------------:|:-----------------:|:-----------------:|
> |        Bilinear       |        59.56       |        59.53       |       59.41      |       59.54       |       59.15       |
> |         JAFAR         |        59.65       |        61.39       |       61.38      |       60.86       |       60.79       |
>
> | **VOC** (mIoU) | 56 ($\times 1.75$) | 112 ($\times 3.5$) | 224 ($\times 7$) | 448 ($\times 14$) | 896 ($\times 28$) |
> |:--------------:|:------------------:|:------------------:|:----------------:|:-----------------:|:-----------------:|
> |    Bilinear    |        80.64       |        80.72       |       80.77      |       80.83       |       80.82       |
> |      JAFAR     |        81.51       |        82.53       |       82.67      |       83.55       |       83.24       |
>
>
> **6. I am curious whether large upsampling ratios will cause Query-Key misalignment or sparse attention issues.**
>
> Across our experiments, we did not observe any noticeable query–key misalignment or sparse attention issues, even at high upsampling ratios. As noted above, JAFAR demonstrates strong robustness under extreme upsampling conditions. To further support this, we will include additional visualizations in the supplementary material that show upsampled feature maps at various resolutions, highlighting the consistency and stability of the attention mechanism across scales.
>
> **7. Also, is RoPE more stable under large-scale resolutions compared to learnable positional bias or sinusoidal encoding?**
>
> In this work, we did not investigate alternative positional embeddings. Following the rationale presented in [5], we adopted RoPE due to its ease of integration, absence of positional artefacts, and strong transferability across spatial resolutions.
>
>
> [1] Liu, Ze, et al. "Swin transformer: Hierarchical vision transformer using shifted windows." Proceedings of the IEEE/CVF international conference on computer vision. 2021.
>
> [2] Hassani, Ali, et al. "Neighborhood attention transformer." Proceedings of the IEEE/CVF conference on computer vision and pattern recognition. 2023.
>
> [3] Pan, Xuran, et al. "Slide-transformer: Hierarchical vision transformer with local self-attention." Proceedings of the IEEE/CVF conference on computer vision and pattern recognition. 2023.
>
> [4] Fan, Qihang, et al. "Rmt: Retentive networks meet vision transformers." Proceedings of the IEEE/CVF conference on computer vision and pattern recognition. 2024.
>
> [5] Darcet, Timothée, et al. "Cluster and predict latent patches for improved masked image modeling." arXiv preprint arXiv:2502.08769 (2025).

---

> > ### Comment · Reviewer_RrSa · 2025-08-05
> >
> > Thank you for the clarification. The author's response has addressed my concerns, and I have no further questions.

---

> > > ### Author Response · Authors · 2025-08-05
> > >
> > > We’re glad to hear that our response has addressed all the concerns raised. If there are no further issues, we would like to kindly invite the reviewer to consider updating their score. Thank you again for your thoughtful and constructive feedback.

---

### Official Review · Reviewer_AAhc · 2025-06-21

**Clarity:** 3
**Significance:** 3
**Originality:** 2
**Rating:** 4
**Confidence:** 4

**Summary:**

This paper presents JAFAR, a light and trainable feature upsampler that can upsample visual features from any foundational vision encoder to an arbitrary target resolution. JAFAR employs an attention-based module to promote semantic alignment between high-resolution queries and semantically enriched low-resolution keys using Spatial Feature Transform (SFT) modulation. This paper also demonstrates that training JAFAR with low upsampling ratios and resolutions can generalize well to significantly higher output scales. Experiments show JAFAR can recover fine-grained spatial details and consistently outperforms existing feature upsampling methods across a diverse set of downstream tasks.

**Questions:**

Referring to the Weaknesses section, I have the following questions:

1. Why does global interpolation perform better in upsampling tasks? Could other parametrized upsamplers also work effectively within the JAFAR framework?

2. What's the real-world use case of high-resolution JAFAR features (*e.g.*, at original image size)? Can JAFAR be integrated into end-to-end training or finetuning pipelines? What's the overall additional computational cost in these cases?

I will adjust my rating according to the author's response.

**Ethical Concerns:**

["NO or VERY MINOR ethics concerns only"]

**Final Justification:**

The authors have addressed my major concerns during the rebuttal.

**Limitations:**

Limitations are briefly discussed in the Conclusion section.

**Quality:**

3

**Strengths And Weaknesses:**

### **Strengths**

1. The writing is smooth and easy to follow.
2. The proposed JAFAR is both architecture and task-agnostic, which can be easily applied to various vision foundation models and downstream tasks.
3. Comprehensive qualitative comparisons illustrate JAFAR's superior performance over other upsampling methods.

### **Weaknesses**

1. **The contribution of this paper is unclear.** This paper follows FeatUp [1] to train a model and task-agnostic framework for feature upsampling to any resolution. Key components such as feature consistency loss and similarity-based upsampling have been well-explored in prior works [2, 3], and the major contribution is the global interpolation mechanism using cross-attention. However, the motivation and necessity of using **global interpolation** for feature upsampling is unclear, since local structures are generally more important for upsampling while considering only local structures can also help reduce computational costs.
2. **Concern of practicality for stand-alone feature upsamplers**. While JAFAR demonstrates improved performance over other (training-free, task-dependent and task-agnostic) upsampling approaches in linear probing settings, it's often impractical to use high-resolution features directly in downstream tasks. More evaluations and analysis beyond the linear probing settings are necessary. What's the advantage of stand-alone feature upsamplers over integrated upsampling modules?

Minor remark: I suggest that the author carefully review the equations in Section 3.1 to ensure that the vector dimensions are consistent for multiplication (Equation (1)), and use different notations for matrix multiplication and element-wise operations (Equations (1) and (2)).



[1] Fu, Stephanie, et al. "FeatUp: A Model-Agnostic Framework for Features at Any Resolution." *The Twelfth International Conference on Learning Representations*.

[2] Yang, Jiawei, et al. "Denoising vision transformers." *European Conference on Computer Vision*. Cham: Springer Nature Switzerland, 2024.

[3] Lu, Hao, et al. "SAPA: Similarity-aware point affiliation for feature upsampling." *Advances in Neural Information Processing Systems* 35 (2022): 20889-20901.

---

> ### Author Rebuttal · Authors · 2025-07-28
>
> We thank the reviewer for the helpful comments and appreciate the positive feedback. Please note that, for the purpose of the rebuttal, all probing layers of the following tables were trained for 10 epochs (instead of 20, as reported in the main paper) to accelerate training while preserving fair comparison.
>
> **Weaknesses**
>
> **1. The contribution of this paper is unclear. This paper follows FeatUp [1] to train a model and task-agnostic framework for feature upsampling to any resolution.**
>
> We would like to clarify that FeatUp does not support feature upsampling to arbitrary resolutions. As noted in lines 97–101, the implicit variant of FeatUp can upsample feature maps only to the original image resolution and requires training a separate upsampler for each image, limiting its generality. The JBU variant, in contrast, stacks $k$ JBU upsampling blocks and is thus limited to a fixed upsampling factor equal to a $2^k$. Our method, however, is the only baseline that supports continuous, task-agnostic feature upsampling to any intermediate resolution between the original feature map and the image resolution.
>
> **2. Key components [...] computational costs.**
>
> We would like to clarify both the role of the attention mechanism in our method and the contributions beyond prior work.
>
> **Cross-attention for continuous-resolution upsampling:**
>
> While similarity-based upsampling has been explored in prior works such as [1], our use of a cross-attention mechanism enables direct control over the output resolution through the spatial dimensions of the queries, determined by the pooling mechanism. This allows feature upsampling not only to the original image resolution but also to arbitrary intermediate resolutions in a task-agnostic manner, a capability not demonstrated by previous methods.
>
> **Importance of SFT:**
>
> As demonstrated in Tab 5, a key contribution of our method is the integration of the SFT module, which softly injects semantic information into the keys. This mechanism enhances spatial resolution while preserving important semantic content from the low-resolution features.
>
> **Architectural flexibility and generalization:**
>
> Thanks to the pooling mechanisms, we are able to train the model at low upsampling ratios and resolutions, while achieving strong generalization to significantly higher upsampling ratios at inference time, demonstrating the robustness and flexibility of our proposed architecture. This robust behavior represents a key empirical contribution of our work.
>
> Regarding the global nature of the cross-attention mechanism, we agree that attending to all keys is not strictly necessary. Attention variants with more localized receptive fields [2, 3, 4], may offer efficiency gains and could be explored in future work. That said, we deliberately avoid overly local mechanisms, as feature maps produced by vision encoders often exhibit spatial misalignments or positional artefacts. Granting each query a larger receptive field via global attention helps mitigate these inconsistencies and improves upsampling quality. Conceptually, this could be related to non-local means in image denoising, where a pixel is refined using information from a broader neighborhood rather than just immediate surroundings.
>
> **3. Concern of practicality for stand-alone feature upsamplers. [...]**
>
> Please Refer to **Question 3** and **Question 4** below.
>
> **4. Minor remark**
>
> We thank the reviewer for pointing this out. We will revise Section 3.1 to ensure that all tensor dimensions are consistent and clearly defined in each equation. In particular, we will explicitly indicate the reshaping of query and key tensors before the dot-product in Equation (1) and clarify the distinction between matrix multiplication and element-wise operations by using a $\odot$ symbol to denote element-wise operation in equation 2).
>
> **Questions**
>
> **1. Why does global interpolation perform better in upsampling tasks?**
>
> Feature maps produced by vision encoders often suffer from imperfections such as spatial misalignments (semantics does not align cleanly with object boundaries), and positional artefacts. Local interpolation methods such as bilinear upsampling are insufficient to correct these issues, as they lack the capacity to incorporate broader contextual information and simply propagate these inconsistencies to higher resolutions without correcting them. In contrast, our use of global attention gives each query access to a wide receptive field, allowing it to integrate information from semantically similar regions across the feature map. This helps mitigate spatial inconsistencies and enhances upsampling quality.
>
> **2. Could other parametrized upsamplers also work effectively within the JAFAR framework?**
>
> A key strength of JAFAR lies in the flexibility of its architectural design, which enables a clear decoupling between training and inference, a feature not commonly found in many existing upsampling approaches. Specifically, the use of adaptive pooling modules plays a dual role:
> - During training, these modules allow supervision at low resolution, thereby avoiding the need for high-res ground truth features, which are typically unavailable.
> - At inference, they support generalization to unseen upsampling factors by adjusting the spatial resolution dynamically.
>
> Existing upsamplers lack this level of flexibility. For example, FeatUp cannot dynamically adjust its output resolution and must be explicitly trained to generate high-res features, followed by learning a separate downsampler to match low-resolution supervision. LiFT, on the other hand, follows a training pipeline similar to JAFAR’s, but its CNN-based architecture inherently limits its ability to generalize to unseen upsampling factors.
>
> In response to the reviewer’s request, and to demonstrate that JAFAR outperforms baselines within the same training framework, we train them to perform $\times 2$ upsampling and evaluate them in a linear probing segmentation setup, with a subsequent bilinear upsampling applied at inference.
>
> | **Cityscapes** (mIoU) | 56    | 112   | 224   |
> |-----------------------|-------|-------|-------|
> | JAFAR                 | 18.68 | 33.72 | 52.02 |
> | LiFT                  | 18.60 | 31.74 | 47.57 |
> | FeatUp                | 18.48 | 31.67 | 48.29 |
> | SAPA                  | 10.20 | 23.16 | 36.96 |
>
> | **VOC** (mIoU) | 56    | 112   | 224   |
> |----------------|-------|-------|-------|
> | JAFAR          | 35.73 | 68.81 | 79.90 |
> | LiFT           | 31.57 | 64.33 | 73.69 |
> | FeatUp         | 33.44 | 64.23 | 75.1  |
> | SAPA           | 20.14 | 53.72 | 69.29 |
>
> **3 What's the real-world use case of high-resolution JAFAR features [...] ?**
>
> High-res features have clear real-world utility in zero-shot dense prediction tasks, such as open-vocabulary semantic segmentation and object localization. Several recent works [5, 6, 7] rely on dense visual features from frozen vision encoders. To match the resolution of the input image, these methods often require an upsampling step, and each currently adopts its own upsampling strategy (e.g., bilinear upsampling, stride modifications, or multi-crop inference). JAFAR offers a simple, effective, and task-agnostic alternative that generalizes across resolutions and tasks. We believe it could serve as a unified upsampling module for these and future zero-shot pipelines, improving both performance and design simplicity. Furthermore, many end-to-end pipelines leverage high-res features for improved performance [8, 9].
>
> **4. Can JAFAR be integrated into end-to-end training or finetuning pipelines? What's the overall additional computational cost in these cases?**
>
> We would like to highlight Section 4.3.5 (Table 4) of the paper, where JAFAR is integrated into three different BeV end-to-end vehicle segmentation pipelines. In these experiments, JAFAR is trained jointly with the rest of the architecture and consistently outperforms the baselines. This demonstrates that JAFAR can be effectively incorporated into end-to-end training or fine-tuning workflows, leading to clear performance improvements. In summary, JAFAR is both practical and versatile: it can function as a plug-and-play component in zero-shot pipelines or be integrated into end-to-end training setups, making it a valuable contribution for a broad range of vision applications.
>
> [1] Lu, Hao, et al. "SAPA: Similarity-aware point affiliation for feature upsampling." Advances in Neural Information Processing Systems 35 (2022): 20889-20901.
>
> [2] Liu, Ze, et al. "Swin transformer: Hierarchical vision transformer using shifted windows." Proceedings of the IEEE/CVF international conference on computer vision. 2021.
>
> [3] Hassani, Ali, et al. "Neighborhood attention transformer." Proceedings of the IEEE/CVF conference on computer vision and pattern recognition. 2023.
>
> [4] Pan, Xuran, et al. "Slide-transformer: Hierarchical vision transformer with local self-attention." Proceedings of the IEEE/CVF conference on computer vision and pattern recognition. 2023.
>
> [5] Mukhoti, Jishnu, et al. "Open vocabulary semantic segmentation with patch aligned contrastive learning." Proceedings of the IEEE/CVF Conference on Computer Vision and Pattern Recognition. 2023.
>
> [6] Couairon, Paul, et al. "Diffcut: Catalyzing zero-shot semantic segmentation with diffusion features and recursive normalized cut." Advances in Neural Information Processing Systems 37 (2024): 13548-13578.
>
> [7] Jose, Cijo, et al. "Dinov2 meets text: A unified framework for image-and pixel-level vision-language alignment." Proceedings of the Computer Vision and Pattern Recognition Conference. 2025.
>
> [8] Li, Kaiyu, et al. "Segearth-ov: Towards training-free open-vocabulary segmentation for remote sensing images." Proceedings of the Computer Vision and Pattern Recognition Conference. 2025.
>
> [9] Zhu, Junzhe et al. “DenseMatcher: Learning 3D Semantic Correspondence for Category-Level Manipulation from a Single Demo.” ArXiv abs/2412.05268 (2024): n. pag.

---

> > ### Comment · Reviewer_AAhc · 2025-08-01
> >
> > Thank the authors for providing clarifications for the contributions of JAFAR, and I agree with the authors that JAFAR offers a more flexible upsampling framework compared to previous works.
> >
> > However, regarding the practical usage for high-resolution features, could the authors elaborate more on the experimental details of BeV end-to-end vehicle segmentation pipelines? For example, what's the resolution after upsampling, and what's the memory usage and computational costs (FLOPs) in these cases?

---

> ### Author Response · Authors · 2025-08-01
>
> In the BeV end-to-end vehicle segmentation pipeline, features are upsampled using JAFAR with a $\times 2$ spatial upsampling factor. The input images are of size $224 \times 476$ (note: there is a typo in the paper which states $496 \times 224$). This results in feature maps with a spatial resolution of $16 \times 34$. Since each scene includes six camera views, the resulting tensor processed by the upsampler has a shape of $(6, 128, 16, 34)$. JAFAR then upsamples this tensor to $(6, 128, 32, 68)$ before it is passed to the downstream BeV processing modules. We report below the GPU memory usage and GFLOPs during training for different upsampling methods. For reference, we include LoftUp as the most recent feature upsampling baseline.
>
> - **SimpleBeV**:
>    - Bilinear: 4981 MiB
>    - FeatUp: 5305 MiB
>    - LoftUp: 5305 MiB
>    - JAFAR: 5453 MiB
>
> - **PointBeV**:
>    - Bilinear: 1833 MiB
>    - FeatUp: 1961 MiB
>    - LoftUp: 2397 MiB
>    - JAFAR: 2521 MiB
>
> - **BeVFormer**:
>    - Bilinear: 2489 MiB
>    - FeatUp: 2639 MiB
>    - LoftUp:  2971MiB
>    - JAFAR: 3135 MiB
>
> Regarding computational cost, the FLOPs required by each method are as follows:
> - **FeatUp**: 0.63 GFLOPs
> - **JAFAR**: 17.12 GFLOPs
> - **LoftUp**: 26.03 GFLOPs
>
> For JAFAR, this corresponds to roughly 4.6% of the total computational cost for SimpleBeV and BeVFormer ($\sim 375$ GFLOPs) and 10% for PointBeV ($\sim 171$ GFLOPs). We will include these elements in the supplementary material for completeness.

---

> ### Comment · Reviewer_AAhc · 2025-08-01
>
> Thank the authors for providing more details. However, I would like to see comparisons of the **overall** memory usage and computational cost compared to baselines (*i.e.*, without upsampling), which can better verify the effectiveness of incorporating JAFAR into end-to-end training pipelines beyond zero-shot and linear probing settings. What dataset do you use for BeV vehicle segmentation? I'm also curious whether JAFAR can outperform the baselines when the vision backbone is unfrozen.

---

> > ### Author Response · Authors · 2025-08-02
> >
> > We apologize for any earlier confusion. The reported memory usage and computational cost refer to total resource consumption (in FP32 precision) during training, including both forward and backward passes, using a batch size of 1. All methods are trained and evaluated on the nuScenes dataset, which includes six camera views per scene. The resulting input shape is $(B, N, C, H, W)$, and in our experiments becomes $(1, 6, 3, 224, 476)$.
> >
> > **SimpleBeV**:
> >
> > |                       | No Upsampling |    Bilinear    |     FeatUp     |      JAFAR     |      LoftUp     |
> > |:---------------------:|:-------------:|:--------------:|:--------------:|:--------------:|:---------------:|
> > | Training Memory (MiB) |      4539     |  4981 (+9.7%)  |  5305 (+16.9%) |  5453 (+20.1%) |  5305 (+16.9%)  |
> > |         GFLOPs        |     594.00    | 595.10 (+0.2%) | 595.73 (+0.3%) | 612.24 (+3.1%) | 621.06 (+ 4.6%) |
> > |          IoU          |     31.75     |      33.67     |      33.95     |      36.59     |        -        |
> >
> > **PointBeV**:
> >
> > |                       | No Upsampling |     Bilinear    |     FeatUp     |      JAFAR      |      LoftUp     |
> > |:---------------------:|:-------------:|:---------------:|:--------------:|:---------------:|:---------------:|
> > | Training Memory (MiB) | 1507          | 1833 (+21.6%)   | 1961 (+30.1%)  | 2521 (+67.3%)   | 2397 (+59.1%)   |
> > |         GFLOPs        | 154.30        | 154.32 (+ 0.0%) | 154.90 (+0.4%) | 171.41 (+11.1%) | 180.25 (+16.8%) |
> > |          IoU          | 34.89         | 36.01           | 35.38          | 37.20           | –               |
> >
> > **BeVFormer**:
> >
> > |                       | No Upsampling |    Bilinear    |      FeatUp     |      JAFAR     |     LoftUp     |
> > |:---------------------:|:-------------:|:--------------:|:---------------:|:--------------:|:--------------:|
> > | Training Memory (MiB) | 2143          | 2489 (+16.1%)  | 2639 (+23.1%)   | 3135 (+46.3%)  | 2971 (+38.6%)  |
> > |         GFLOPs        | 352.40        | 354.70 (+0.6%) | 357.72 (+ 1.5%) | 376.74 (+6.9%) | 379.82 (+7.8%) |
> > |          IoU          | 33.72         | 34.18          | 34.01           | 36.54          | –              |
> >
> >
> > While JAFAR introduces additional overhead compared to baselines without upsampling, this increase remains within practical limits. As shown in Table 4 and the results above, the added computational and memory cost is accompanied by clear and consistent performance gains (IoU) across all settings. Furthermore, JAFAR is fully compatible with bfloat16, enabling mixed-precision training and inference, which helps reduce memory footprint in resource-constrained environments.
> >
> > In Section 4.3.5, we followed the common practice of using a frozen foundation vision backbone, as is frequently done in both general end-to-end pipelines [1, 2, 3] and in the Bird’s Eye View literature [4, 5]. We agree that unfreezing the backbone is a valuable direction that could further validate the effectiveness of JAFAR. We appreciate this suggestion, recognize its importance, and intend to explore it in future work.
> >
> > [1] Xu, Jiarui, et al. "Open-vocabulary panoptic segmentation with text-to-image diffusion models." Proceedings of the IEEE/CVF conference on computer vision and pattern recognition. 2023.
> >
> > [2] Yu, Qihang, et al. "Convolutions die hard: Open-vocabulary segmentation with single frozen convolutional clip." Advances in Neural Information Processing Systems 36 (2023): 32215-32234.
> >
> > [3] Yang, Lihe, et al. "Depth anything v2." Advances in Neural Information Processing Systems 37 (2024): 21875-21911.
> >
> > [4] Barın, Merve Rabia, Görkay Aydemir, and Fatma Güney. "Robust Bird's Eye View Segmentation by Adapting DINOv2." arXiv preprint arXiv:2409.10228 (2024).
> >
> > [5] Sophia Sirko-Galouchenko et al. “OccFeat: Self-supervised Occupancy Feature Prediction”, IEEE/CVF Conference on Computer Vision and Pattern Recognition Workshops (CVPRW), 2022.

---

> > > ### Comment · Reviewer_AAhc · 2025-08-04
> > >
> > > Thank the authors for the clarification and for providing detailed information regarding memory usage and computational cost. However, it is important to note that previous studies such as the Depth Anything series [1, 2] and OccFeat [3] utilize frozen DINOv2 features for feature alignment but still update the entire prediction network in an end-to-end manner. Moreover, as reported in Table 2 of the paper [4] that the authors mentioned, freezing the backbone network significantly reduces performance (34.3 mIoU) compared to full-rank fine-tuning (41.5 mIoU) or low-rank fine-tuning (42.3 mIoU). This confirms the necessity of evaluating the performance with a tunable backbone to ensure practical applicability in real-world scenarios.
> > >
> > > Furthermore, as reported by the authors, using stand-alone upsamplers even with small upsampling factors (*e.g.*, 2) results in a notable increase in memory consumption for downstream networks. Thus, the practicability of any-resolution feature upsamplers appears limited in this case. It seems more promising to integrate JAFAR into existing networks directly by replacing the upsampling operations (*e.g.*, FeatUP [5] variant of Segformer).
> > >
> > > While I appreciate the authors' contribution compared to previous works, my concern hasn't been adequately addressed. Besides, I cannot find the information regarding the dataset used for the BeV experiment in the paper. Could the authors please provide further clarification on these points?
> > >
> > >
> > >
> > >
> > > [1] Yang, Lihe, et al. "Depth anything: Unleashing the power of large-scale unlabeled data." *Proceedings of the IEEE/CVF conference on computer vision and pattern recognition*. 2024.
> > >
> > > [2] Yang, Lihe, et al. "Depth anything v2." *Advances in Neural Information Processing Systems* 37 (2024): 21875-21911.
> > >
> > > [3] Sirko-Galouchenko, Sophia, et al. "Occfeat: Self-supervised occupancy feature prediction for pretraining bev segmentation networks." *Proceedings of the IEEE/CVF Conference on Computer Vision and Pattern Recognition*. 2024.
> > >
> > > [4] Barın, Merve Rabia, Görkay Aydemir, and Fatma Güney. "Robust Bird's Eye View Segmentation by Adapting DINOv2." *arXiv preprint arXiv:2409.10228* (2024).
> > >
> > > [5] Fu, Stephanie, et al. "Featup: A model-agnostic framework for features at any resolution." *arXiv preprint arXiv:2403.10516* (2024).

---

> ### Author Response · Authors · 2025-08-06
>
> We thank the reviewer for their interest in our work and for keeping the discussion open. Before addressing the specific questions, we would like to briefly remind the core objective of our work.
>
> We demonstrate through this paper that JAFAR leads to **significant performance improvements** across several tasks with frozen foundation vision encoders: linear probing for semantic segmentation and depth estimation (Table 1), zero-shot open-vocabulary segmentation (Table 3), and Bird’s-Eye-View (BeV) segmentation (Table 4).
>
> We would like to emphasize that the BeV segmentation experiment is not intended to achieve state-of-the-art performance on the task itself. Rather, it serves to test JAFAR’s **robustness and effectiveness** within a more complex pipeline than linear probing. The frozen backbone setup used in this experiment is consistent with the rest of our paper and aligned with prior work in the literature [1, 2], where similar setups are used. Within this setup, JAFAR outperforms baseline upsampling methods, providing further validation of its benefits. Please note that fine-tuning the backbone for the BEV task is not compatible with the rebuttal timeline, but integrating JAFAR into the pipeline is both feasible and a promising future work.
>
> In response to the request, we demonstrate below that JAFAR can also enhance **end-to-end trained models**. Specifically, we integrate JAFAR into the **EoMT [3]** (CVPR 2025) segmentation framework by replacing the original upsampling module (based on transposed convolutions) with JAFAR. This yields a 3-point performance gain, while maintaining a reasonable overhead in terms of memory and FLOPs. Importantly, we note that **FeatUp**, which stacks four JBU blocks (×16 upsampling), incurs **higher memory overhead** while performing worse than the baseline. As mentioned earlier, future work will focus on incorporating **more local attention mechanisms** to further improve computational efficiency. We would be glad to include these new results in the revised version of the paper, demonstrating that JAFAR can be successfully included in an end-to-end training pipeline with backbone fine-tuning.
>
> | Cityscapes: 224 | Baseline | JAFAR | FeatUp |
> |:---------------:|:--------:|:-----:|:------:|
> |   Memory (GiB)  |   2 467  | 2 677 | 11 465 |
> |      GFLOPs     |    179   |  463  |   496  |
> |       mIoU      |   60.5   |  63.5 |  60.0  |
>
> **About nuScenes**
>
> nuScenes is a widely used dataset for autonomous driving research [4]. We apologize for the omission and will add the relevant details to the paper. The dataset comprises 1,000 driving scenes, each lasting 20 seconds, captured using six synchronized cameras, resulting in a total of 1.4 million images.
>
> Finally, we agree that supporting arbitrary feature resolutions does not imply that image-resolution features are always necessary. However, one of the strengths of JAFAR lies in its **flexibility**, it allows the upsampling factor to be adjusted dynamically based on the requirements of the task, whether integrated into a fully end-to-end pipeline or used in zero-shot settings. In practice, we argue that many downstream tasks **do benefit from high-resolution features**, and having the ability to produce them directly can be highly valuable in real-world applications.
>
> We sincerely hope these new results address your concerns and help support a more favorable assessment of our submission.
>
>
> [1] Xu, Jiarui, et al. "Open-vocabulary panoptic segmentation with text-to-image diffusion models." Proceedings of the IEEE/CVF conference on computer vision and pattern recognition. 2023.
>
> [2] Yu, Qihang, et al. "Convolutions die hard: Open-vocabulary segmentation with single frozen convolutional clip." Advances in Neural Information Processing Systems 36 (2023): 32215-32234.
>
> [3] Kerssies, Tommie, et al. "Your vit is secretly an image segmentation model." Proceedings of the Computer Vision and Pattern Recognition Conference. 2025.
>
> [4] Caesar, Holger, et al. "nuscenes: A multimodal dataset for autonomous driving." Proceedings of the IEEE/CVF conference on computer vision and pattern recognition. 2020.

---

> > ### Comment · Reviewer_AAhc · 2025-08-06
> >
> > I would like to thank the authors for providing clarifications and additional experimental results. Based on the reported performance and cost analysis with the EoMT framework, I believe my major concern has been addressed. I suggest the authors include detailed experimental information regarding BEV and the results of integrating JAFAR with the EoMT segmentation framework, either within the main body or in the appendix of the final manuscript.
> >
> > After rebuttal, I will adjust my rating to Borderline accept and would encourage future comprehensive investigation of high-resolution features beyond the presented scenarios.

---

### Official Review · Reviewer_ne7u · 2025-06-29

**Clarity:** 3
**Significance:** 3
**Originality:** 3
**Rating:** 5
**Confidence:** 4

**Summary:**

In this paper, the authors propose a lightweight and flexible feature upsampler named JAFAR. It employs an attention-based structure to upsample features from any base visual encoder to arbitrary output resolution. The authors evaluate the performance of JAFAR on a diverse set of downstream tasks.

**Questions:**

See weaknesses.

**Ethical Concerns:**

["NO or VERY MINOR ethics concerns only"]

**Final Justification:**

The authors have fully addressed my concerns during the author response phase. Therefore, I have raised my rating from 4 to 5.

**Limitations:**

The paper lacks an analysis of the limitations of its research.

**Quality:**

3

**Strengths And Weaknesses:**

**Strengths:**

(1) The authors propose a task-independent, lightweight, and training-friendly feature upsampling method that can enhance visual features at any resolution, breaking through the limitations of existing methods in resolution, generalization, and interpretability.

(2) The authors compare multiple existing upsampling methods and conducted empirical comparisons in multiple tasks (classification, depth estimation, semantic segmentation) and at different resolutions, demonstrating the performance advantages of JAFAR.

(3) JAFAR achieve a tradeoff between parameter quantity and inference time, with only 0.7M parameters and low computational cost at most output resolutions, making it suitable for practical deployment.

(4) The paper is well-written and well-organized.

**Weaknesses:**

(1) Although the method works well, the authors lack theoretical derivation and explanation of the JAFAR upsampling mechanism, and do not clearly explain why its structure can improve spatial resolution while retaining semantic information.

(2) The paper's method can achieve different upsampling ratios. Will model performance deteriorate as the upsampling ratio increases?

(3) Can the paper's method be further extended to image super-resolution tasks?

---

> ### Author Rebuttal · Authors · 2025-07-28
>
> We thank the reviewer for the helpful comments and appreciate the positive feedback. Please note that, for the purpose of the rebuttal, all probing layers of the following tables were trained for 10 epochs (instead of 20, as reported in the main paper) to accelerate training while preserving fair comparison.
>
>
> **1. Although the method works well, the authors lack theoretical derivation and explanation of the JAFAR upsampling mechanism, and do not clearly explain why its structure can improve spatial resolution while retaining semantic information.**
>
> While our method is primarily empirical, its core design shares conceptual similarities with non-local means filtering used in image denoising. In non-local means, each pixel is denoised by aggregating information from all other pixels in the image, weighted by their similarity to the target pixel, rather than relying solely on local neighborhoods, as in traditional filters. This principle of leveraging semantically similar but spatially distant information is directly applicable to our setting and helps explain the effectiveness of our approach. We will include this analysis in the revised version of the paper.
>
> In the context of feature upsampling, this is particularly useful because low-resolution feature maps produced by vision encoders often suffer from spatial misalignments and positional artefacts. Traditional local upsampling methods like bilinear interpolation simply propagate these inconsistencies to higher resolutions without correcting them.
>
> JAFAR addresses this limitation by learning a similarity-based weighting scheme via a cross-attention mechanism, where high-resolution queries, derived from the original image, attend to low-resolution keys from the same image encoding, semantically enriched through the SFT module. As shown in Table 5, this design greatly enhances the semantic relevance of the learned similarities. In contrast, the **Linear Projection** baseline, where queries are taken from a shallow image encoding and keys directly from the semantic feature map, performs noticeably worse. This is likely due to the mismatch between the two representation spaces, which hinders the quality of the computed similarities. Our proposed mechanism enables each high-resolution query location to dynamically gather information from semantically aligned key regions, effectively mimicking non-local means in a more flexible, learned fashion.
>
> To further highlight the value of the **SFT mechanism**, we will include the following line in Table 5 demonstrating the performance gain from semantic feature injection. We will re-run the experiment with 20 epochs to include it in the main paper.
>
> |               | Cityscapes | VOC   | ADE20K |
> |---------------|------------|-------|--------|
> | JAFAR         | 60.86      | 83.55 | 41.04  |
> | JAFAR wo/ SFT | 56.76      | 82.54 | 40.27  |
>
> **2. The paper's method can achieve different upsampling ratios. Will model performance deteriorate as the upsampling ratio increases?**
>
> To evaluate the robustness of JAFAR across varying upsampling ratios, we conducted an experiment (shown in the tables below) assessing its performance at multiple scales, including extreme ratios. The results demonstrate that JAFAR maintains strong performance even as the upsampling ratio increases, confirming its ability to generalize well beyond the training range. We will include these tables in the supplementary material to further support the method’s robustness and scalability.
>
> **Input Feature Size: 8**
>
> | **Cityscapes** (mIoU) | 56 ($\times 7$) | 112 ($\times 14$) | 224 ($\times 28$) | 448 ($\times 56$) | 896 ($\times 112$) |
> |:---------------------:|:---------------:|:-----------------:|:-----------------:|:-----------------:|:------------------:|
> |        Bilinear       |      31.87      |       31.81       |       31.90       |       31.61       |        31.92       |
> |         JAFAR         |      36.54      |       36.04       |       35.71       |       35.77       |        35.54       |
>
> | **VOC** (mIoU) | 56 ($\times 7$) | 112 ($\times 14$) | 224 ($\times 28$) | 448 ($\times 56$) | 896 ($\times 112$) |
> |:--------------:|:---------------:|:-----------------:|:-----------------:|:-----------------:|:------------------:|
> |    Bilinear    |      64.79      |       64.95       |       64.89       |       64.91       |        64.9        |
> |      JAFAR     |       72.7      |       73.13       |       72.77       |       72.69       |        72.55       |
>
>
> **Input Feature Size: 16**
>
> | **Cityscapes** (mIoU) | 56 ($\times 3.5$) | 112 ($\times 7$) | 224 ($\times 14$) | 448 ($\times 28$) | 896 ($\times 56$) |
> |:---------------------:|:-----------------:|:----------------:|:-----------------:|:-----------------:|:-----------------:|
> |        Bilinear       |       47.77       |       47.77      |       47.78       |       47.83       |       47.84       |
> |         JAFAR         |       53.18       |       53.22      |       52.35       |       51.76       |       51.36       |
>
> | **VOC** (mIoU) | 56 ($\times 3.5$) | 112 ($\times 7$) | 224 ($\times 14$) | 448 ($\times 28$) | 896 ($\times 56$) |
> |:--------------:|:-----------------:|:----------------:|:-----------------:|:-----------------:|:-----------------:|
> |    Bilinear    |       75.41       |       75.42      |       75.51       |       75.50       |       75.50       |
> |      JAFAR     |       80.57       |       80.76      |       80.58       |       80.36       |       80.24       |
>
>
> **Input Feature Size: 32**
>
> | **Cityscapes** (mIoU) | 56 ($\times 1.75$) | 112 ($\times 3.5$) | 224 ($\times 7$) | 448 ($\times 14$) | 896 ($\times 28$) |
> |:---------------------:|:------------------:|:------------------:|:----------------:|:-----------------:|:-----------------:|
> |        Bilinear       |        59.56       |        59.53       |       59.41      |       59.54       |       59.15       |
> |         JAFAR         |        59.65       |        61.39       |       61.38      |       60.86       |       60.79       |
>
> | **VOC** (mIoU) | 56 ($\times 1.75$) | 112 ($\times 3.5$) | 224 ($\times 7$) | 448 ($\times 14$) | 896 ($\times 28$) |
> |:--------------:|:------------------:|:------------------:|:----------------:|:-----------------:|:-----------------:|
> |    Bilinear    |        80.64       |        80.72       |       80.77      |       80.83       |       80.82       |
> |      JAFAR     |        81.51       |        82.53       |       82.67      |       83.55       |       83.24       |
>
>
> **3.  Can the paper's method be further extended to image super-resolution tasks?**
>
> We have not applied our method to image super-resolution tasks, so we cannot confidently assess its effectiveness in that domain. However, upsampling latent representations produced by super-resolution pipelines that operate in a latent space [1] might be feasible. For example, we could think of applying JAFAR to the latents of a pre-trained VAE, then decoding them to obtain higher-resolution images. That said, this adaptation would likely require significant modifications, including the use of more localized cross-attention mechanisms and the introduction of image-specific loss terms. Given these considerations, we view this as a non-trivial but promising direction for future research.
>
> **4. The paper lacks an analysis of the limitations of its research.**
>
> Thank you for pointing this out. We briefly mention some limitations in the conclusion but we will expand on the existing discussion in the main text and include a dedicated limitations section in the supplementary material. This section will specifically address the computational cost and necessity of global versus local cross-attention mechanisms, incorporating the points raised during the rebuttal.
>
> [1]Chen, Yinbo, et al. "Image neural field diffusion models." Proceedings of the IEEE/CVF Conference on Computer Vision and Pattern Recognition. 2024.

---

> > ### Comment · Reviewer_ne7u · 2025-08-05
> >
> > Thanks to the authors for providing a detailed response. My comments have been adequately addressed.

---

> > > ### Author Response · Authors · 2025-08-05
> > >
> > > We’re glad to hear that our response has addressed all the concerns raised. If there are no further issues, we would like to kindly invite the reviewer to consider updating their score. Thank you again for your thoughtful and constructive feedback.

---

> > > > ### Comment · Reviewer_ne7u · 2025-08-06
> > > >
> > > > I have raised my rating from 4 to 5.

---

### Official Review · Reviewer_hVFB · 2025-07-02

**Clarity:** 3
**Significance:** 3
**Originality:** 3
**Rating:** 4
**Confidence:** 4

**Summary:**

The paper propose a lightweight and flexible feature upsampler designed to enhance the spatial resolution of visual features from any Foundation Vision Encoder to any target resolution. JAFAR features an attention-based upsampling module that aligns the spatial representations of high-resolution queries with semantically enriched low-resolution keys via Spatial Feature Transform modulation.

**Questions:**

see weakness

**Ethical Concerns:**

["NO or VERY MINOR ethics concerns only"]

**Final Justification:**

My concerns are addressed. I keep my positive score.

**Quality:**

3

**Strengths And Weaknesses:**

Strengths:

This paper proposes an interesting approach that enables adaptive super-resolution for arbitrary resolutions. The use of RoPE to generalize to unseen resolutions, combined with attention mechanisms and zero-shot training, is quite innovative.

Compared to previous methods, the proposed approach not only addresses the challenge of arbitrary-resolution upsampling but also achieves superior performance.

Weaknesses:

I would like to see an ablation study to evaluate whether RoPE indeed improves the model's generalization ability to different spatial resolutions.

The concept of zero-shot super-resolution has been explored as early as 2017 in the SR community (e.g., ZSSR [1]). While not entirely new, applying this idea to a new super-resolution task can still be considered a modest contribution.

[1] “Zero-Shot” Super-Resolution using Deep Internal Learning

---

> ### Author Rebuttal · Authors · 2025-07-28
>
> We thank the reviewer for the helpful comments and appreciate the positive feedback. Please note that, for the purpose of the rebuttal, all probing layers of the following tables were trained for 10 epochs (instead of 20, as reported in the main paper) to accelerate training while preserving fair comparison.
>
> **1. I would like to see an ablation study to evaluate whether RoPE indeed improves the model's generalization ability to different spatial resolutions.**
>
> In response to the reviewer’s recommendation, we provide the following ablation study to evaluate the impact of RoPE on JAFAR’s generalization ability in semantic segmentation probing across multiple spatial resolutions $(56, 112, 224, 448)$. To ensure consistency, the upsampling factor is fixed ($\times 14$) across all resolutions.
>
> | **Cityscapes** (mIoU)  | 56    | 112   | 224   | 448   |
> |----------------|-------|-------|-------|-------|
> | JAFAR w/ RoPE  | 21.41 | 35.8  | 52.42 | 60.86 |
> | JAFAR wo/ RoPE | 18.15 | 27.06 | 34.80 | 36.09 |
>
>
> |     **VOC**  (mIoU)   |   56  |  112  |  224  |  448  |
> |:--------------:|:-----:|:-----:|:-----:|:-----:|
> |  JAFAR w/ RoPE | 40.41 | 72.78 | 80.57 | 83.55 |
> | JAFAR wo/ RoPE | 35.36 | 62.38 | 67.58 | 67.22 |
>
> This ablation study highlights that RoPE plays a critical role in maintaining consistent spatial alignment between queries and keys across varying spatial resolutions, with its benefits becoming more pronounced at higher resolutions. We will include this ablation in the supplementary material to underscore the critical role of RoPE in enabling effective generalization to varying spatial resolutions.
>
>
> **2. The concept of zero-shot super-resolution has been explored as early as 2017 in the SR community (e.g., ZSSR [1]). While not entirely new, applying this idea to a new super-resolution task can still be considered a modest contribution.**
>
> While we acknowledge that our approach shares a conceptual similarity with ZSSR in image super-resolution (i.e using a downsampled version of the target to predict the target) we emphasize several important differences, through which we introduce what we believe are significant contributions.
>
> A key distinction lies in the nature of the target: while high-resolution images are easy to obtain, acquiring high-resolution feature maps is significantly more challenging. While ZSSR typically downsamples images by modest factors, modern vision encoders reduce spatial resolution much more aggressively, by factors of $14$, $16$, or even $32$. This discrepancy severely limits the range of scaling factors that can be simulated during training, making it much harder to match the large upsampling ratios needed at inference time.
> One might consider increasing input image resolution to produce higher-resolution feature maps. However, this approach introduces substantial problems: high-resolution inputs push the vision encoder far outside its training distribution, and although resizing positional embeddings can compensate to some degree, it quickly reaches its limits (l.34 [1, 2]). We evaluated this approach as the **Large Image** baseline in Tables 1–3 and Figure 4, which consistently underperforms compared to the Bilinear Upsampling baseline.
>
> Despite these limitations, our experiments (Tables 1–4, Figure 4) consistently show that training the proposed cross-attention architecture at low upsampling ratios (between $\times 2$ and $\times 4$) and resolutions leads to strong generalization at much higher scales ($\times 14$). This robust behavior, both surprising and significant, represents a key empirical contribution of our work.
>
>
> [1] Yang, Jiawei, et al. "Denoising vision transformers." European Conference on Computer Vision. Cham: Springer Nature Switzerland, 2024.
>
> [2] Fan, Qihang et al. “ViTAR: Vision Transformer with Any Resolution.” ArXiv abs/2403.18361 (2024): n. pag.

---

> > ### Author Response · Authors · 2025-08-05
> >
> > Thank you again for the insightful feedback. As the discussion period has been extended, we are happy to continue the dialogue and address any remaining concerns. Please let us know if there is anything further we can clarify, and we will respond promptly.

---

### Decision · Program_Chairs · 2025-09-17

**Decision:**

Accept (poster)

**Comment:**

This paper received overall positive reviews. Reviewers recognized it as an interesting approach to enable adaptive super-resolution for arbitrary resolutions, validated across multiple tasks. The rebuttal effectively addressed the initial concerns, resulting in a version suitable for acceptance. The AC concurs with the majority and considers this a solid contribution. It is recommended that the clarifications and supporting evidence from the rebuttal be incorporated into the camera-ready version. Looking forward to seeing this work presented at NeurIPS.